# Discriminative reconstruction via simultaneous dense and sparse coding

**Abiy Tasissa**  *abiy.tasissa@tufts.edu*
*Department of Mathematics*
*Tufts University*

**Emmanouil Theodosis**  *etheodosis@g.harvard.edu*
*School of Engineering and Applied Sciences*
*Harvard University*

**Bahareh Tolooshams**  *btoloosh@caltech.edu*
*Computing + Mathematical Sciences*
*California Institute of Technology*

**Demba Ba**  *demba@seas.harvard.edu*
*School of Engineering and Applied Sciences*
*Harvard University*

**Reviewed on OpenReview:** *https://openreview.net/forum?id=FkgM06HEbk*

## Abstract

Discriminative features extracted from the sparse coding model have been shown to perform well for classification. Recent deep learning architectures have further improved reconstruction in inverse problems by considering new dense priors learned from data. We propose a novel dense and sparse coding model that integrates both representation capability and discriminative features. The model studies the problem of recovering a dense vector $\mathbf{x}$ and a sparse vector $\mathbf{u}$ given measurements of the form $\mathbf{y} = \mathbf{Ax} + \mathbf{Bu}$. Our first analysis relies on a geometric condition, specifically the minimal angle between the spanning subspaces of matrices $\mathbf{A}$ and $\mathbf{B}$, which ensures a unique solution to the model. The second analysis shows that, under some conditions on $\mathbf{A}$ and $\mathbf{B}$, a convex program recovers the dense and sparse components. We validate the effectiveness of the model on simulated data and propose a dense and sparse autoencoder (DenSaE) tailored to learning the dictionaries from the dense and sparse model. We demonstrate that (i) DenSaE denoises natural images better than architectures derived from the sparse coding model ($\mathbf{Bu}$), (ii) in the presence of noise, training the biases in the latter amounts to implicitly learning the $\mathbf{Ax} + \mathbf{Bu}$ model, (iii) $\mathbf{A}$ and $\mathbf{B}$ capture low- and high-frequency contents, respectively, and (iv) compared to the sparse coding model, DenSaE offers a balance between discriminative power and representation.

## 1 Introduction

The problem of representing data as a linear mixture of components finds applications in many areas of signal and image processing (Bertalmio et al., 2003; Mallat & Yu, 2010). Variational partial differential equations (PDE) approaches in Aujol et al. (2003) and Vese & Osher (2003) develop an image decomposition model whereby a target image is decomposed into a piecewise smooth component (cartoon) and a periodic component (texture). Another approach posits that a desirable model should use multiple matrices, each capturing different structures in the image; for example, $\mathbf{\Phi}_1$ is a matrix modeling texture and $\mathbf{\Phi}_2$ is a matrix modeling cartoon. Along these methods is Morpohological Component Analysis (MCA) (Elad et al., 2005) which utilizes pre-specified dictionaries and a a sparsity prior on the representation coefficients. In the

simplest setting, an image $\mathbf{y}$ is modeled in MCA as $\mathbf{y} \approx \sum_{k=1}^{2} \boldsymbol{\Phi}_k \mathbf{x}_k$ where $\boldsymbol{\Phi}_1, \boldsymbol{\Phi}_2$ are the pre-specified structured dictionaries capturing different morphologies of the image. The MCA model is then based on the idea that the sparsest representation of each component is attained in the pre-set dictionary component. In contrast to MCA which imposes sparsity for all mixture coefficients, the proposed model in this paper adopts a combination of smooth and sparse regularization.

Related motivating problems for representing data as a linear mixture are the background/foreground separation problem (Zhou et al., 2012) and anomaly detection in images (Chang & Chiang, 2002). In the former problem, a widely used approach is the robust principal component analysis (RPCA) algorithm (Candès et al., 2011) which is based on decomposing a data matrix into low rank (background) and sparse (foreground) components. The latter problem, anomaly detection with images, arises in most manufacturing processes and methods based on decomposing the images into smooth and sparse components have been studied (Yan et al., 2017; Shen et al., 2022); however, to the best of the authors' knowledge, no theoretical analysis has been conducted for that decomposition. In addition, the sparse and smooth components in image anomaly detection are localized, inhibiting the generality of the model.

In this paper, we study a model which decomposes a signal or image in consideration into a smooth part, a sparse part, and noise. The prototypical form of our model is $\mathbf{y} = \mathbf{A}\mathbf{x} + \mathbf{B}\mathbf{u} + \mathbf{e}$, where $\mathbf{y} \in \mathbb{R}^m$ is the measured signal, $\mathbf{A}\mathbf{x}$ is the smooth component, $\mathbf{B}\mathbf{u}$ is the sparse component and $\mathbf{e}$ is the noise. The smooth component is generated from the dictionary $\mathbf{A} \in \mathbb{R}^{m \times p}$ using a smooth coefficient $\mathbf{x}$. In turn, the sparse component is generated from the dictionary $\mathbf{B} \in \mathbb{R}^{m \times n}$ using a sparse vector $\mathbf{u}$. We use Tikhonov regularization (Tikhonov et al., 1995) and $\ell_1$ regularization to promote smoothness and sparsity, respectively. With that, the general non-noisy problem we study in this paper is

$$\min_{\mathbf{x} \in \mathbb{R}^p, \mathbf{u} \in \mathbb{R}^n} \quad ||\mathbf{G}\mathbf{x}||_2^2 + ||\mathbf{u}||_1 \quad \text{subject to} \quad \mathbf{y} = \mathbf{A}\mathbf{x} + \mathbf{B}\mathbf{u}, \tag{1}$$

where $\mathbf{G}$ is the Tikhonov regularization operator. Here on, we refer to the model in (1) as the dense and sparse coding problem. The word dense here refers to the smooth part and is used to emphasize that we do not require any sparsity.

Both MCA and the dense and sparse coding problem naturally lend themselves to and share a common theme with sparse coding and dictionary learning. Sparsity regularized deep neural network architectures have been used for image denoising and discriminative tasks such as image classification (Simon & Elad, 2019; Tolooshams et al., 2020; Rolfe & LeCun, 2013). Recent work has highlighted some limitations of convolutional sparse coding (CSC) autoencoders and its multi-layer and deep generalizations (Sulam et al., 2019; Zazo et al., 2019) for data reconstruction. The work in Simon & Elad (2019) argues that the sparsity levels that CSC allows can only accommodate very sparse vectors, making it unsuitable to capture all features of signals such as natural images, and propose to compute the minimum mean-squared error solution under the CSC model, which is a dense vector capturing a richer set of features. Moreover, the majority of CSC frameworks subtract low-frequency components of images prior to applying convolutional dictionary learning (CDL) for representation purposes (Garcia-Cardona & Wohlberg, 2018); however, this is not desired for denoising tasks where noise corrupts low frequencies in addition to the high spectrum. The goals of the paper are twofold. First, it presents a theoretical analysis of the dense and sparse coding problem. Second, we argue the dense representation $\mathbf{x}$ in a dictionary $\mathbf{A}$ is useful for reconstruction and the sparse representation $\mathbf{u}$ has discriminative capability.

## 1.1 Organization of paper

**Section 2** discusses related work. In **Section 3**, we give some technical background to the main analysis. **Section 4** presents the theoretical analysis of the paper. Phase transition, classification, and denoising experiments appear in **Section 5**. We conclude in **Section 6**. We start by defining notations.

## 1.2 Notation

Lowercase and uppercase boldface letters denote column vectors and matrices, respectively. Given a vector $\mathbf{x} \in \mathbb{R}^n$ and a support set $S \subset \{1, ..., n\}$, $\mathbf{x}_S$ denotes the restriction of $\mathbf{x}$ to indices in $S$. $\text{supp}(\mathbf{x})$ denotes

the support of a vector, defined as $\text{supp}(x) = \{i : x_i \neq 0\}$. Given a set $S$, $S^c$ denotes its complement. For a matrix $\mathbf{A} \in \mathbb{R}^{m \times p}$, $\mathbf{A}_S$ is a submatrix of size $m \times |S|$ with column indices in $S$. The column space of a matrix $\mathbf{A}$ is designated by $\text{Col}(\mathbf{A})$, its null space by $\text{Ker}(\mathbf{A})$. We denote the Euclidean, $\ell_0$, $\ell_1$, and $\ell_\infty$ norms of a vector $\mathbf{x}$, respectively as $||\mathbf{x}||_2$, $||\mathbf{x}||_0$, $||\mathbf{x}||_1$, and $||\mathbf{x}||_\infty$. The operator and infinity norm of a matrix $\mathbf{A}$ are respectively denoted as $||\mathbf{A}||$ and $||\mathbf{A}||_\infty$. The sign function, applied componentwise to a vector $\mathbf{x}$, is denoted by $\text{sgn}(\mathbf{x})$. The indicator function is denoted by $\mathbb{1}$. The column vector $\mathbf{e}_i$ denotes the vector of zeros except a 1 at the $i$-th location. The orthogonal complement of a subspace $\boldsymbol{W}$ is denoted by $\boldsymbol{W}^\perp$. The operator $\mathcal{P}_{\boldsymbol{W}}$ denotes the orthogonal projection operator onto the subspace $\boldsymbol{W}$. $\log(x)$ denotes the logarithm of $x$ in base $e$. $\oplus$ denotes the direct sum of subspaces. $E(X)$ denotes the expectation of a random variable $X$.

## 2 Related work

### 2.1 Compressive sensing from union of dictionaries

The dense and sparse coding problem is similar in flavor to sparse recovery in the union of dictionaries (Donoho & Stark, 1989; Donoho & Huo, 2001; Elad & Bruckstein, 2002; Donoho & Elad, 2003; Kuppinger et al., 2011). Consider two sets of orthonormal bases, $\mathbf{A} \in \mathbb{R}^{m \times m}$ and $\mathbf{B} \in \mathbb{R}^{m \times m}$, in $\mathbb{R}^m$. One setting is based on the idea that a given signal will have a sparse representation with respect to suitably predefined $\mathbf{A}$ or $\mathbf{B}$ (Mallat, 1999; Daubechies, 1992; Dobson & Santosa, 1996). The concept of the union of bases is based on constructing an overcomplete dictionary $[\mathbf{A} \ \mathbf{B}]$ from $\mathbf{A}$ and $\mathbf{B}$. This model then posits that a signal $\mathbf{y}$ can be represented as $\mathbf{y} = \mathbf{Ax} + \mathbf{Bu}$, where $\mathbf{x}$ and $\mathbf{u}$ are sparse. Notably, the signal may not be individually sparse with respect to $\mathbf{A}$ and $\mathbf{B}$, but it has sparsity in the joint representation (Elad & Bruckstein, 2002; Donoho & Huo, 2001). In Donoho & Stark (1989), the authors apply this model to error-correcting encryption and the separation of two signals. In both cases, the measured signal is assumed to be a superposition of two components, each sparse in a predefined dictionary. Most results in the literature of union of bases take the form of an uncertainty principle that relates the sum of the sparsity of $\mathbf{x}$ and $\mathbf{u}$ to the mutual coherence between $\mathbf{A}$ and $\mathbf{B}$, and which guarantees that the representation is unique and identifiable by $\ell_1$ minimization. In Donoho & Stark (1989), the authors study the problem of recovering $\mathbf{x}$ and $\mathbf{u}$ from $\mathbf{y} = \mathbf{Ax} + \mathbf{Bu}$, where $\mathbf{A} \in \mathbb{R}^{m \times m}$ is a discrete Fourier matrix (DFT) and $\mathbf{B} \in \mathbb{R}^{m \times m}$ is the identity matrix. Therein, under the assumption that the support of $\mathbf{u}$ is known, the result shows that $\mathbf{x}$ can be exactly recovered if $2||\mathbf{x}||_0 ||\mathbf{u}||_0 < m$. This result for the Fourier-identity pair was further generalized to the case where $\mathbf{A}$ and $\mathbf{B}$ are orthonormal bases (Elad & Bruckstein, 2002) and when the measurements come from a union of non-orthogonal bases (Donoho & Elad, 2003; Kuppinger et al., 2011). We remark that all the aforementioned results assume sparsity on both $\mathbf{x}$ and $\mathbf{u}$.

### 2.2 Error correction

The problem of recovering a signal $\mathbf{x}$ given the measurement model $\mathbf{y} = \mathbf{Ax} + \mathbf{u}$, where $\mathbf{u}$ is a sparse error vector, is known as the sparse error correction problem (Candes et al., 2005). The work therein considers a tall measurement matrix $\mathbf{A}$ and assumes that the fraction of corrupted entries is suitably bounded. To obtain a sparse minimization program, the matrix $\mathbf{A}$ is eliminated by a matrix $\mathbf{B}$, where $\mathbf{BA} = \mathbf{0}$, resulting $\mathbf{By} = \mathbf{Bu}$. The resulting model is then solved via $\ell_1$ minimization and exactness is shown under the restricted isometry condition on $\mathbf{B}$. The works in Wright & Ma (2010); Nguyen & Tran (2013); Pope et al. (2013); Studer et al. (2011); Studer & Baraniuk (2014) study the general error correction problem $\mathbf{y} = \mathbf{Ax} + \mathbf{Bu}$ under different models of the signal $\mathbf{x}$, the sparse interference vector $\mathbf{u}$ and the matrices $\mathbf{A}$ and $\mathbf{B}$.

We note that all the aforementioned works consider a single sparsifying norm to recover the components $\mathbf{x}$ and $\mathbf{u}$. In contrast, we use mixed norms that impose the Tikhonov and sparsity regularization. We also note that when $\mathbf{A}$ has more columns than rows, one of the settings we consider in this paper, our problem departs from the sparse error correction problem.

### 2.3 Weighted Lasso

When there is some prior on the support of the underlying sparse vector $\mathbf{u}$ given measurements of the form $\mathbf{y} = \mathbf{Bu}$, the works in Lian et al. (2018); Mansour & Saab (2017); Vaswani & Lu (2010); Oymak et al. (2012);

Friedlander et al. (2011); Flinth (2016) consider a weighted Lasso program. Assume that the estimate of the support of $\mathbf{u}$ is $T \subset \{1, ..., n\}$. The standard weighted Lasso considers the following optimization program

$$\min_{\mathbf{u}} \quad ||\mathbf{u}||_{1,w} \quad \text{subject to} \quad \mathbf{y} = \mathbf{Bu},$$

where $||\mathbf{u}||_{1,\boldsymbol{w}} = \sum_{i=1}^{n} w_i u_i$ and $\boldsymbol{w} \in \mathbb{R}^n$ indicates the vector of weights. For instance, Mansour & Saab (2017) set the weights as follows: $w_i \in [0, 1]$ if $i \in T$ and 1 otherwise. The aforementioned works discuss the recovery of the underlying signal using weighted Lasso under some conditions on the weights, size and accuracy of the estimated support. One approach for the feasibility constraint in (1) is to reformulate it as a weighted Lasso problem in a combined dictionary $[\mathbf{A}\ \mathbf{B}]$ where the support estimate is the support of $\mathbf{x}$. This formulation has few limitations. First, the recovery guarantees in weighted Lasso require accuracy of the estimated support which would depend on the number of columns of $\mathbf{A}$. Second, these guarantees impose uniform structure on the combined dictionary, such as the weighted null space conditions, whereas our conditions allow a deterministic dictionary $\mathbf{A}$ and a random dictionary $\mathbf{B}$. Finally, even in the regime where weighted Lasso might succeed in recovering the underlying $\mathbf{x}$ and $\mathbf{u}$, it does not guarantee that $\mathbf{Ax}$ is smooth as that regularization is not reflected in the Lasso optimization.

## 2.4 Morphological component analysis (MCA)

Morphological Component Analysis (MCA) considers the decomposition of a signal or image into different components with each component having a specific structure (morphology) (Starck et al., 2004; 2005; Bobin et al., 2007; Elad et al., 2005). In MCA, the specific structure is imposed via sparsity with respect to pre-specified dictionaries. Specifically, given $K$ dictionaries $\{\boldsymbol{\Phi}_1, ..., \boldsymbol{\Phi}_K\}$, we model the data $\mathbf{y}$ as a superposition of $K$ linear components as follows: $\mathbf{y} = \sum_{i=1}^{K} \boldsymbol{\Phi}_i \mathbf{y}_i$ where $\mathbf{y}_i$ is assumed to be sparse in $\boldsymbol{\Phi}_i$ but it is not as sparse (or not sparse at all) in other dictionaries. With this set up, the dictionaries discriminate between the different components. In Starck et al. (2004) a basis pursuit approach to solve the MCA is employed and its utility is demonstrated in examples such as separating texture from the smooth part of an image. The main difference of our model from MCA is a smooth-sparse decomposition as opposed to sparse-sparse decomposition with pre-set dictionaries that contain information about targeted morphologies. The theory and analysis of the two models is also markedly different as we consider a smoothness regularizer.

## 2.5 Dictionary learning and unrolling

Given a data set, learning a dictionary in which each example admits a sparse representation is useful in a number of tasks (Aharon et al., 2006; Mairal et al., 2011). This problem, known as sparse coding (Olshausen & Field, 1997) or dictionary learning (Agarwal et al., 2016; Garcia-Cardona & Wohlberg, 2018), has been the subject of investigation in recent years in the signal processing community. A growing body of work, referred to as algorithm unrolling (Monga et al., 2021), has mapped the sparse coding problem into encoders for sparse recovery (Gregor & Lecun, 2010). In this paper, we unroll the dense and sparse coding problem for a principled design of an autoencoder. The designed autoencoder efficiently learns a dense representation $\mathbf{x}$, useful for reconstruction, and a sparse representation $\mathbf{u}$ with discriminative capability.

## 2.6 Contribution

We focus on (1) and start by first providing theoretical guarantees for the feasibility problem $\mathbf{y} = \mathbf{Ax} + \mathbf{Bu}$. Our first result is based on a geometric condition on the minimum principal angle between certain subspaces. Next, we prove that the convex program in (1) recovers the underlying components $\mathbf{x}$ and $\mathbf{u}$ under some assumptions on the measurement matrices and the Tikhonov regularizer. We empirically validate the effectiveness of our methods through phase transition curves and make a direct comparison to noisy compressive sensing, highlighting the latter's inability to recover signals adhering to our proposed model. We then apply our optimization algorithm to sense real data and validate the expected properties of the sparse and smooth components.

We connect the proposed model to dictionary learning/algorithm unrolling by proposing a dense and sparse autoencoder (DenSaE). We demonstrate the superior discriminative and representation capabilities of DenSaE

compared to sparse coding networks in the task of reconstruction and classifying MNIST dataset. In this task, we characterize the role of dense and sparse learned components; we argue that the dense representation $\mathbf{x}$ is useful for reconstruction and the sparse representation $\mathbf{u}$ has discriminative capability. Moreover, for image denoising, we show that DenSaE outperforms other networks which are based only on sparse coding.

## 3  Technical background

We start by briefly reviewing uniqueness results for compressive sensing with exact measurements. The typical assumption in these analysis is to assume the measurement matrices satisfy certain conditions such as the restricted isometry property (RIP) and coherence. In our setting, we employ RIPless recovery analysis (Candes & Plan, 2011; Kueng & Gross, 2014), which will be referred to and used in the proof of Theorem 6. We note that existing anisotropic analysis is based on a single measurement matrix. For our model, the anisotropic analysis is applied to a certain matrix derived from $\mathbf{A}$, $\mathbf{B}$ and the Tikhonov regularizer $\mathbf{G}$.

### 3.1  Compressive sensing with exact measurements

An underdetermined linear system $\mathbf{y} = \mathbf{Bu}$ where $\mathbf{y} \in \mathbb{R}^m$ and $\mathbf{B} \in \mathbb{R}^{m \times n}$ with $n > m$ generally has infinitely many solutions. One prior for the recovery of the solution is sparsity of the underlying vector $\mathbf{u}$. This leads to the following optimization problem

$$\min_{\mathbf{u} \in \mathbb{R}^n} \quad ||\mathbf{u}||_0 \quad \text{subject to} \quad \mathbf{y} = \mathbf{Bu}. \tag{2}$$

While (2) is a natural program, it is known to be computationally intractable. Common alternatives are basis pursuit (BP) (Tropp, 2004; Chen et al., 2001) and iterative greedy methods such as orthogonal matching pursuit (OMP)(Davis et al., 1997; Pati et al., 1993; Tropp & Gilbert, 2007). In basis pursuit, (2) is modified by considering a convex relaxation of the $\ell_0$ norm leading to the $\ell_1$ minimization problem.

$$\min_{\mathbf{u} \in \mathbb{R}^n} \quad ||\mathbf{u}||_1 \quad \text{subject to} \quad \mathbf{y} = \mathbf{Bu}. \tag{3}$$

A fundamental question is under what conditions the above optimization program identifies the underlying solution for (2). A related part of this question is when the $\ell_0$ minimization problem admits a unique solution. One condition is the restricted isometry property (RIP) (Candes & Tao, 2005). The $s$-restricted isometry constant of an $m \times n$ measurement matrix $\mathbf{B}$ is the smallest constant $\delta_s$ such that

$$(1 - \delta_s)||\mathbf{u}||^2 \leq ||\mathbf{Bu}||^2 \leq (1 + \delta_s)||\mathbf{u}||^2,$$

holds for all $s$-sparse signals $\mathbf{u}$ (an $s$-sparse vector has at most $s$ non-zero entries). Small RIP constants ensure unique solution of the $\ell_0$ minimization problem and further provide the guarantee that the $\ell_1$ relaxation in (3) is exact (Candes, 2008). It is known that random measurement matrices have small RIP constants (Candes & Tao, 2006; Baraniuk et al., 2008). Another condition for analysis is based on the coherence of a measurement matrix $\mathbf{B}$ defined as

$$\mu = \max_{i \neq j} |\mathbf{b}_i^T \mathbf{b}_j|,$$

where $\mathbf{b}_i$ denotes the $i$-th column of $\mathbf{B}$ which is assumed to be unit-norm. To state the next result, consider $\mathbf{y} = \mathbf{Bu}^*$, where $\mathbf{u}^*$ is the underlying sparse vector. If $||\mathbf{u}^*||_0 < \frac{1}{2}\left(1 + \frac{1}{\mu}\right)$, the $\ell_0$ problem admits a unique solution and both BP and OMP relaxations are exact (Donoho & Elad, 2003; Gribonval & Nielsen, 2003; Tropp, 2004). We note that coherence conditions are typically stronger than those based on the restricted isometry property, specially for random matrices.

### 3.2  Compressive sensing with structured random matrices

We review the technical results in Candes & Plan (2011); Kueng & Gross (2014) which consider conditions for the exact recovery of the $\ell_1$ minimization problem for the setting of structured measurement matrices. In this setting, we do not assume restricted isometry property (RIP) of the measurement matrices (Candes & Tao,

2005) and this broadens the sensing matrices that can be employed. We consider a sequence of i.i.d. random vectors $\mathbf{b}_1, ..., \mathbf{b}_m$ drawn from some distribution $F$ on $\mathbb{R}^n$ and with measurements defined as $\mathbf{y}_i = \mathbf{b}_i^T \mathbf{u}^*$. Two properties are essential to the analysis. The first is the *isotropy property* which states that $E[\mathbf{b}\mathbf{b}^T] = \mathbf{I}$ for $\mathbf{b} \sim F$. The second property is the *incoherence property*. The incoherence parameter is the smallest number $\mu(F)$ such that

$$\max_{1 \leq i \leq n} |\langle \mathbf{b}, \mathbf{e}_i \rangle|^2 \leq \mu(F), \tag{4}$$

holds almost surely for any $\mathbf{b} \sim F$. Given these assumptions, the work in Candes & Plan (2011) provides the following theoretical result.

**Theorem 1** (Candes & Plan (2011))**.** *Let* $\mathbf{B} = \frac{1}{\sqrt{m}} \sum_{i=1}^m \mathbf{e}_i \mathbf{b}_i^T$ *be a measurement matrix and let* $\mathbf{u}$ *be a fixed but otherwise arbitrary s-sparse vector in* $\mathbb{R}^n$. *Then with probability at least* $1 - \frac{5}{n} - e^{-\beta}$ *for some positive constant* $\beta$, $\mathbf{u}$ *is the unique minimizer to*

$$\min_{\mathbf{u} \in \mathbb{R}^n} \quad ||\mathbf{u}||_1 \quad subject \ to \quad \mathbf{y} = \mathbf{B}\mathbf{u},$$

*provided that* $m \geq C_\beta \mu(F) s \log n$. *The constant* $C_\beta = C_0(1 + \beta)$ *where* $C_0$ *is some constant.*

We note that the definition of $\mathbf{B}$ based on re-scaling the vectors $\mathbf{b}_1, \mathbf{b}_2, ..., \mathbf{b}_m$ is done so that the columns of $\mathbf{B}$ are approximately unit-normed. The work in Kueng & Gross (2014) extends the above result without assuming the isotropy property. Let $\boldsymbol{\Sigma} = E[\mathbf{b}\mathbf{b}^T]^{\frac{1}{2}}$, where as before $\mathbf{b}$ is drawn from distribution $F$ on $\mathbb{R}^n$, denote the covariance matrix. The anisotropic analysis in Kueng & Gross (2014) is based on two properties pertaining to the measurement matrix. The first is the incoherence property which is formally defined as follows. The incoherence parameter is the smallest number $\mu(F)$ such that

$$\max_{1 \leq i \leq n} |\langle \mathbf{b}, \mathbf{e}_i \rangle|^2 \leq \mu(F) \quad \text{and} \quad \max_{1 \leq i \leq n} |\langle \mathbf{b}, E[\mathbf{b}\mathbf{b}^*]^{-1} \mathbf{e}_i \rangle|^2 \leq \mu(F), \tag{5}$$

hold almost surely. The second property is based on the conditioning of the covariance matrix. Specifically, it is based on an *s*-sparse condition number, for which we restate the definition from Kueng & Gross (2014).

**Definition 1** (Kueng & Gross (2014))**.** *The largest and smallest s-sparse eigenvalue of a matrix* $\mathbf{X}$ *are given by* $\lambda_{\max}(s, \mathbf{X}) := \max_{\mathbf{v}: ||\mathbf{v}||_0 \leq s} \frac{||\mathbf{X}\mathbf{v}||_2}{||\mathbf{v}||_2}$ *and* $\lambda_{\min}(s, \mathbf{X}) := \min_{\mathbf{v}: ||\mathbf{v}||_0 \leq s} \frac{||\mathbf{X}\mathbf{v}||_2}{||\mathbf{v}||_2}$. *The s-sparse condition number of* $\mathbf{X}$ *is* $cond(s, \mathbf{X}) = \frac{\lambda_{\max}(s, \mathbf{X})}{\lambda_{\min}(s, \mathbf{X})}$.

Given these assumptions, the main result in Kueng & Gross (2014) is stated below.

**Theorem 2** (Kueng & Gross (2014))**.** *Let* $\mathbf{B} = \frac{1}{\sqrt{m}} \sum_{i=1}^m \mathbf{e}_i \mathbf{b}_i^T$ *be a measurement matrix and let* $\mathbf{u}$ *be a fixed but otherwise arbitrary s-sparse vector in* $\mathbb{R}^n$. *Define* $\kappa_s = \max\{cond(s, \boldsymbol{\Sigma}), cond(s, \boldsymbol{\Sigma}^{-1})\}$ *and let* $\beta \geq 1$. *If the number of measurements fulfills* $m \geq C \kappa_s \mu(F) \beta^2 s \log n$, *then the solution* $\mathbf{u}$ *of the convex program* $\min_{\mathbf{u}} ||\mathbf{u}||_1$ *subject to* $\mathbf{y} = \mathbf{B}\mathbf{u}$, *is unique and equal to* $\mathbf{u}^*$ *with probability at least* $1 - e^{-\beta}$.

The proofs of Theorem 1 and Theorem 2 are based on the dual certificate approach. The approach involves assuming the existence of a dual certificate vector $\mathbf{v}$ that satisfies certain conditions which ensures the uniqueness of the minimization problem. The remaining task is then to construct a dual certificate that satisfies these conditions. Since the conditions on the dual certificate will be used in our main analysis, we summarize the conditions on $\mathbf{v}$ in the following lemma.

**Lemma 1** (Kueng & Gross (2014),Candes & Plan (2011))**.** *Under the assumptions of 1 and Theorem 2, there exists a dual certificate vector* $\mathbf{v}$, *in the row space of* $\mathbf{B}$, *such that*

$$||\mathbf{v}_S - \operatorname{sgn}(\mathbf{u}_S^*)||_2 \leq \tfrac{1}{4} \quad and \quad ||\mathbf{v}_{S^c}||_\infty \leq \tfrac{1}{4}. \tag{6}$$

*In addition, it follows that*

$$||\boldsymbol{\Delta}_S||_2 \leq 2||\boldsymbol{\Delta}_{S^c}||_2, \quad for \ any \quad \boldsymbol{\Delta} \in \operatorname{Ker}(\mathbf{B}). \tag{7}$$

## 4 Theoretical Analysis

The dense and sparse coding problem studies the solutions of the linear system $\mathbf{y} = \mathbf{Ax} + \mathbf{Bu}$. Given matrices $\mathbf{A} \in \mathbb{R}^{m \times p}$ and $\mathbf{B} \in \mathbb{R}^{m \times n}$ and a vector $\mathbf{y} \in \mathbb{R}^m$, the goal is to provide conditions under which there is a unique solution $(\mathbf{x}^*, \mathbf{u}^*)$, where $\mathbf{u}^*$ is $s$-sparse, and an algorithm for recovering it. For ease of navigating the main results, Figure 1 shows the road-map of the main theoretical results.

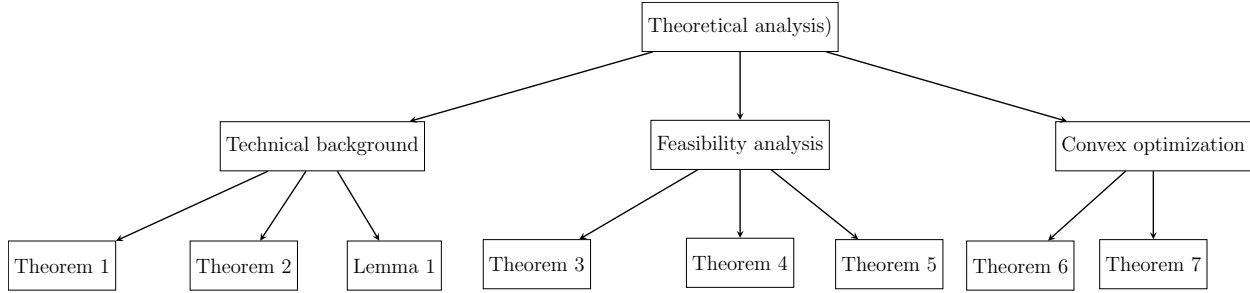

Figure 1: A road-map of the main theoretical results in the manuscript.

### 4.1 Feasibility problem

In this subsection, we first study the uniqueness of solutions to the linear system accounting for the different structures the measurement matrices $\mathbf{A}$ and $\mathbf{B}$ can have. A uniqueness result can be readily obtained by assuming orthogonality of $\mathrm{Col}(\mathbf{A})$ and $\mathrm{Col}(\mathbf{B})$. In what follows, we establish uniqueness results for the general setting. The main result of this subsection is Theorem 4 which, under a natural geometric condition based on the minimum principal angle between the column space of $\mathbf{A}$ and the span of $s$ columns in $\mathbf{B}$, establishes a uniqueness result for the dense and sparse coding problem. Since the vector $\mathbf{u}$ in the proposed model is sparse, we consider the classical setting of an overcomplete measurement matrix $\mathbf{B}$ with $n \gg m$. The next theorem provides a uniqueness result assuming a certain direct sum representation of the space $\mathbb{R}^m$. We first note that, for $\mathbf{A} \in \mathbb{R}^{m \times p}$ with $p \gg m$, any $\boldsymbol{z} \in \mathrm{Ker}(\mathbf{A})$ can be added to any solution $\mathbf{x}$. With that, we consider uniqueness modulo the vectors in $\mathrm{Ker}(\mathbf{A})$, meaning any feasible solution is assumed to be in $\mathrm{Ker}(\mathbf{A})^\perp$.

**Theorem 3.** *Assume that there exists at least one solution to $\mathbf{y} = \mathbf{Ax} + \mathbf{Bu}$, namely the pair $(\mathbf{x}^*, \mathbf{u}^*)$. Let $S$, with $|S| = s$, denote the support of $\mathbf{u}^*$. If $\mathbf{B}_S$ has full column rank and $\mathbb{R}^m = \mathrm{Col}(\mathbf{A}) \oplus \mathrm{Col}(\mathbf{B}_S)$, the only unique solution to the linear system, with the condition that any feasible $s$-sparse vector $\mathbf{u}$ is supported on $S$ and any feasible $\mathbf{x}$ is in $\mathrm{Ker}(\mathbf{A})^\perp$, is $(\mathbf{x}^*, \mathbf{u}^*)$.*

*Proof.* Let $(\mathbf{x}, \mathbf{u})$, with $\mathbf{u}$ supported on $S$ and $\mathbf{x} \in \mathrm{Ker}(\mathbf{A})^\perp$, be another solution pair. It follows that $\mathbf{A}\boldsymbol{\delta_1} + \mathbf{B}_S(\boldsymbol{\delta_2})_S = \mathbf{0}$ where $\boldsymbol{\delta_1} = \mathbf{x} - \mathbf{x}^*$ and $\boldsymbol{\delta_2} = \mathbf{u} - \mathbf{u}^*$. Let $\mathbf{U} \in \mathbb{R}^{m \times r}$ and $\mathbf{V} \in \mathbb{R}^{m \times q}$ be matrices whose columns are the orthonormal bases of $\mathrm{Col}(\mathbf{A})$ and $\mathrm{Col}(\mathbf{B}_S)$, respectively. The equation $\mathbf{A}\boldsymbol{\delta_1} + \mathbf{B}_S(\boldsymbol{\delta_2})_S = \mathbf{0}$ can equivalently be written as $\sum_{i=1}^{r}\langle \mathbf{A}\boldsymbol{\delta_1}, \mathbf{U}_i\rangle \mathbf{U}_i + \sum_{i=1}^{q}\langle \mathbf{B}_S(\boldsymbol{\delta_2})_S, \mathbf{V}_i\rangle \mathbf{V}_i = \mathbf{0}$ with $\mathbf{U}_i$ and $\mathbf{V}_i$ denoting the $i$-th columns of $\mathbf{U}$ and $\mathbf{V}$, respectively. More compactly, we have $\begin{bmatrix} \mathbf{U} & \mathbf{V} \end{bmatrix} \begin{bmatrix} \{\langle \mathbf{A}\boldsymbol{\delta_1}, \mathbf{U}_i\rangle\}_{i=1}^{r} \\ \{\langle \mathbf{B}_S(\boldsymbol{\delta_2})_S, \mathbf{V}_i\rangle\}_{i=1}^{q} \end{bmatrix} = \mathbf{0}$. Noting that the matrix $\begin{bmatrix} \mathbf{U} & \mathbf{V} \end{bmatrix}$ has full column rank, the homogeneous problem admits the trivial solution implying that $\mathbf{A}\boldsymbol{\delta_1} = \mathbf{0}$ and $\mathbf{B}_S(\boldsymbol{\delta_2})_S = \mathbf{0}$. Since $\mathbf{B}_S$ has full column rank and $\boldsymbol{\delta_1} \in \{\mathrm{Ker}(\mathbf{A}) \cap \mathrm{Ker}(\mathbf{A})^\perp\}$, it follows that $\boldsymbol{\delta_1} = \boldsymbol{\delta_2} = \mathbf{0}$. Therefore, $(\mathbf{x}^*, \mathbf{u}^*)$ is the unique solution. $\square$

The uniqueness result in the above theorem hinges on the representation of the space $\mathbb{R}^m$ as the direct sum of the subspaces $\mathrm{Col}(\mathbf{A})$ and $\mathrm{Col}(\mathbf{B}_S)$. We use the definition of the minimal principal angle between two subspaces, and its formulation in terms of singular values (Björck & Golub, 1973), to derive an explicit geometric condition for the uniqueness analysis of the linear system in the general case.

**Definition 2.** *Let $\mathbf{U} \in \mathbb{R}^{m \times r}$ and $\mathbf{V} \in \mathbb{R}^{m \times q}$ be matrices whose columns are the orthonormal basis of $\mathrm{Col}(\mathbf{A})$ and $\mathrm{Col}(\mathbf{B})$, respectively. The minimum principal angle between the subspaces $\mathrm{Col}(\mathbf{A})$ and $\mathrm{Col}(\mathbf{B})$ is defined*

*as follows*

$$\cos(\mu(\mathbf{U}, \mathbf{V})) = \max_{\mathbf{u} \in \mathrm{Col}(\mathbf{U}), \mathbf{v} \in \mathrm{Col}(\mathbf{V})} \frac{\mathbf{u}^T \mathbf{v}}{||\mathbf{u}||_2 ||\mathbf{v}||_2}. \tag{8}$$

*In addition, $\cos(\mu(\mathbf{U}, \mathbf{V})) = \sigma_1(\mathbf{U}^T\mathbf{V})$ where $\sigma_1$ denotes the largest singular value.*

**Theorem 4.** *Assume that there exists at least one solution to $\mathbf{y} = \mathbf{Ax} + \mathbf{Bu}$, namely the pair $(\mathbf{x}^*, \mathbf{u}^*)$. Let $S$, with $|S| = s$, denote the support of $\mathbf{u}^*$. Assume that $\mathbf{B}_S$ has full column rank . Let $\mathbf{U} \in \mathbb{R}^{m \times r}$ and $\mathbf{V} \in \mathbb{R}^{m \times q}$ be matrices whose columns are the orthonormal bases of $\mathrm{Col}(\mathbf{A})$ and $\mathrm{Col}(\mathbf{B}_S)$, respectively. If $\cos(\mu(\mathbf{U}, \mathbf{V})) = \sigma_1(\mathbf{U}^T\mathbf{V}) < 1$, the only unique solution to the linear system, with the condition that any feasible $s$-sparse vector $\mathbf{u}$ is supported on $S$ and any feasible $\mathbf{x}$ is in $\mathrm{Ker}(\mathbf{A})^\perp$, is $(\mathbf{x}^*, \mathbf{u}^*)$.*

*Proof.* Consider any candidate solution pair $(\mathbf{x}^* + \boldsymbol{\delta_1}, \mathbf{u}^* + \boldsymbol{\delta_2})$ with $\boldsymbol{\delta_2}$ supported in S. We will prove uniqueness by showing that $\mathbf{A}\boldsymbol{\delta_1} + \mathbf{B}_S(\boldsymbol{\delta_2})_S = 0$ if and only if $\boldsymbol{\delta_1} = \mathbf{0}$ and $\boldsymbol{\delta_2} = \mathbf{0}$. Using the orthonormal basis set $\mathbf{U}$ and $\mathbf{V}$, $\mathbf{A}\boldsymbol{\delta_1} + \mathbf{B}_S(\boldsymbol{\delta_2})_S$ can be represented as : $\mathbf{A}\boldsymbol{\delta_1} + \mathbf{B}_S(\boldsymbol{\delta_2})_S = \begin{bmatrix} \mathbf{U} & \mathbf{V} \end{bmatrix} \begin{bmatrix} \mathbf{U}^T\mathbf{A}\boldsymbol{\delta_1} \\ \mathbf{V}^T\mathbf{B}_S(\boldsymbol{\delta_2})_S \end{bmatrix}$. For simplicity of notation, let $\mathbf{K}$ denote the block matrix: $\mathbf{K} = \begin{bmatrix} \mathbf{U} & \mathbf{V} \end{bmatrix}$. If we can show that the columns of $\mathbf{K}$ are linearly independent, it follows that $\mathbf{A}\boldsymbol{\delta_1} + \mathbf{B}_S(\boldsymbol{\delta_2})_S = \mathbf{0}$ if and only if $\mathbf{A}\boldsymbol{\delta_1} = \mathbf{0}$ and $\mathbf{B}_S(\boldsymbol{\delta_2})_S = \mathbf{0}$. We now consider the matrix $\mathbf{K}^T\mathbf{K}$ which has the following representation

$$\mathbf{K}^T\mathbf{K} = \begin{bmatrix} [\mathbf{I}]_{r \times r} & [\mathbf{U}^T\mathbf{V}]_{r \times q} \\ [\mathbf{V}^T\mathbf{U}]_{q \times r} & [\mathbf{I}]_{q \times q} \end{bmatrix}$$
$$= \begin{bmatrix} [\mathbf{I}]_{r \times r} & [\mathbf{0}]_{r \times q} \\ [\mathbf{0}]_{q \times r} & [\mathbf{I}]_{q \times q} \end{bmatrix} + \begin{bmatrix} [\mathbf{0}]_{r \times r} & [\mathbf{U}^T\mathbf{V}]_{r \times q} \\ [\mathbf{V}^T\mathbf{U}]_{q \times r} & [\mathbf{0}]_{q \times q} \end{bmatrix}.$$

With the singular value decomposition of $\mathbf{U}^T\mathbf{V}$ being $\mathbf{U}^T\mathbf{V} = \mathbf{Q}\boldsymbol{\Sigma}\mathbf{R}^T$, the last matrix in the above representation has the following equivalent form $\begin{bmatrix} \mathbf{0} & \mathbf{U}^T\mathbf{V} \\ \mathbf{V}^T\mathbf{U} & \mathbf{0} \end{bmatrix} = \begin{bmatrix} \mathbf{Q} & \mathbf{0} \\ \mathbf{0} & \mathbf{R} \end{bmatrix} \begin{bmatrix} \mathbf{0} & \boldsymbol{\Sigma} \\ \boldsymbol{\Sigma} & \mathbf{0} \end{bmatrix} \begin{bmatrix} \mathbf{Q} & \mathbf{0} \\ \mathbf{0} & \mathbf{R} \end{bmatrix}^T$. It now follows that $\begin{bmatrix} \mathbf{0} & \mathbf{U}^T\mathbf{V} \\ \mathbf{V}^T\mathbf{U} & \mathbf{0} \end{bmatrix}$ is *similar* to the matrix $\begin{bmatrix} \mathbf{0} & \boldsymbol{\Sigma} \\ \boldsymbol{\Sigma} & \mathbf{0} \end{bmatrix}$. Hence, the nonzero eigenvalues of $\mathbf{K}^T\mathbf{K}$ are $1 \pm \sigma_i$, $1 \le i \le \min(p, q)$, with $\sigma_i$ denoting the $i$-th largest singular value of $\mathbf{U}^T\mathbf{V}$. Using the assumption $\sigma_1 < 1$ results the bound $\lambda_{\min}(\mathbf{K}^T\mathbf{K}) > 0$. It follows that the columns of $\mathbf{K}$ are linearly independent, and hence $\mathbf{A}\boldsymbol{\delta_1} = \mathbf{0}$ and $\mathbf{B}_S(\boldsymbol{\delta_2})_S = \mathbf{0}$. Since $\mathbf{B}_S$ is full column rank and $\boldsymbol{\delta_1} \in \{\mathrm{Ker}(\mathbf{A}) \cap \mathrm{Ker}(\mathbf{A})^\perp\}$, it follows that $\boldsymbol{\delta_1} = \mathbf{0}$ and $\boldsymbol{\delta_2} = \mathbf{0}$.

$\square$

A restrictive assumption of the above theorem is that the support of the sought-after $s$-sparse solution $\mathbf{u}^*$ is known. We can remove this assumption by considering $\mathrm{Col}(\mathbf{A})$ and $\mathrm{Col}(\mathbf{B}_T)$ where $T$ is an arbitrary subset of $\{1, 2, ..., n\}$ with $|T| = s$. More precisely, we state the following corollary whose proof is similar to the proof of Theorem 4.

**Corollary 1.** *Assume that there exists at least one solution to $\mathbf{y} = \mathbf{Ax} + \mathbf{Bu}$, namely the pair $(\mathbf{x}^*, \mathbf{u}^*)$. Let $S$, with $|S| = s$, denote the support of $\mathbf{u}^*$ and $T$ be an arbitrary subset of $\{1, 2, ..., n\}$ with $|T| \le s$. Assume that any $2s$ columns of $\mathbf{B}$ are linearly independent. Let $\mathbf{U} \in \mathbb{R}^{m \times p}$ and $\mathbf{V} \in \mathbb{R}^{m \times q}$ be matrices whose columns are the orthonormal bases of $\mathrm{Col}(\mathbf{A})$ and $\mathrm{Col}(\mathbf{B}_{S \cup T})$, respectively. If $\mu(\mathbf{U}, \mathbf{V}) = \sigma_1(\mathbf{U}^T\mathbf{V}) < 1$, holds for all choices of $T$, the only unique solution to the linear system is $(\mathbf{x}^*, \mathbf{u}^*)$ with the condition that any feasible $\mathbf{u}$ is $s$-sparse and any feasible $\mathbf{x}$ is in $\mathrm{Ker}(\mathbf{A})^\perp$.*

Of interest is the identification of simple conditions such that $\sigma_1(\mathbf{U}^T\mathbf{V}) < 1$. The following theorem proposes one such condition to establish uniqueness.

**Theorem 5.** *Assume that there exists at least one solution to $\mathbf{y} = \mathbf{Ax} + \mathbf{Bu}$, namely the pair $(\mathbf{x}^*, \mathbf{u}^*)$. Let $S$, with $|S| = s$, denote the support of $\mathbf{u}^*$. Assume that $\mathbf{B}_S$ has full column rank. Let $\mathbf{U} \in \mathbb{R}^{m \times r}$ and $\mathbf{V} \in \mathbb{R}^{m \times q}$ be matrices whose columns are the orthonormal bases of $\mathrm{Col}(\mathbf{A})$ and $\mathrm{Col}(\mathbf{B}_S)$, respectively. Let $\max_{i,j} |\boldsymbol{u}_i^T \boldsymbol{v}_j| = \mu$. If $s < \frac{1}{\sqrt{r}\mu}$, the only unique solution to the linear system, with the condition that any feasible $s$-sparse vector $\mathbf{u}$ is supported on $S$ and any feasible $\mathbf{x}$ is in $\mathrm{Ker}(\mathbf{A})^\perp$, is $(\mathbf{x}^*, \mathbf{u}^*)$.*

*Proof.* It suffices to show that $\sigma_1 < 1$. Noting that $\sigma_1 = ||\mathbf{U}^T\mathbf{V}||_2$, we use the following matrix norm inequality $||\mathbf{U}^T\mathbf{V}||_2 \leq \sqrt{r}||\mathbf{U}^T\mathbf{V}||_\infty$ as follows: $\sigma_1 \leq \sqrt{r}\,||\mathbf{U}^T\mathbf{V}||_\infty \leq \sqrt{r}\mu s < 1$. □

The constant $\mu$ is the coherence of the matrix $\mathbf{U}^T\mathbf{V}$ (Donoho et al., 2005; Tropp, 2004). The above result states that if the mutual coherence of $\mathbf{U}^T\mathbf{V}$ is small, we can accommodate increased sparsity of the underlying signal component $\mathbf{u}^*$. We note that, up to a scaling factor, $\sigma_1(\mathbf{U}^T\mathbf{V})$ is the block coherence of $\mathbf{U}$ and $\mathbf{V}$ (Eldar et al., 2010). However, unlike the condition in Eldar et al. (2010), we do not restrict the dictionaries $\mathbf{A}$ and $\mathbf{B}$ to have linearly independent columns. In the next subsection, we propose and analyze a convex program to recover the dense and sparse vectors.

## 4.2 Dense and sparse recovery via convex optimization

We propose the following convex optimization program for the dense and sparse coding problem

$$\min_{\mathbf{x}\in\mathbb{R}^p, \mathbf{u}\in\mathbb{R}^n} \; ||\mathbf{Gx}||_2^2 + ||\mathbf{u}||_1 \quad \text{subject to} \quad \mathbf{y} = \mathbf{Ax} + \mathbf{Bu}. \tag{9}$$

**Proof strategy:** The Tikhonov matrix $\mathbf{G}$ is assumed to have full column rank. Our goal is to establish that, under certain conditions, the above minimization problem admits a unique solution $(\mathbf{x}^*, \mathbf{u}^*)$. The proof strategy is as follows: First, we make the change of variables $\mathbf{z} = \mathbf{Gx}$, which allows us to rewrite the linear constraint as $\mathbf{y} = \mathbf{Hx} + \mathbf{Bu}$, where $\mathbf{H} = \mathbf{AG}^\dagger$. The second step is to eliminate the dense component in the constraint by applying the projection operator $\mathcal{P}_{\mathrm{Col}(\mathbf{H})^\perp}$. This yields $\mathcal{P}_{\mathrm{Col}(\mathbf{H})^\perp}(\mathbf{y}) = \mathcal{P}_{\mathrm{Col}(\mathbf{H})^\perp}(\mathbf{Bu})$. Denoting $\tilde{\mathbf{C}} = \mathcal{P}_{\mathrm{Col}(\mathbf{H})^\perp}(\mathbf{B})$, we obtain $\mathcal{P}_{\mathrm{Col}(\mathbf{H})^\perp}(\mathbf{y}) = \tilde{\mathbf{C}}\mathbf{u}$. Focusing on the optimization problem with respect to $\mathbf{u}$, we have a sparse recovery problem with the measurement matrix $\tilde{\mathbf{C}}$. To guarantee exact recovery for this problem, certain conditions on the matrix $\tilde{\mathbf{C}}$ are required. For instance, the results discussed in Section 3.2 require the rows of $\tilde{\mathbf{C}}$ to be sampled i.i.d. from some distribution $F$. However, if $\mathbf{A}$ and $\mathbf{B}$ are realized as random i.i.d. matrices, the projection operator does not necessarily preserve the i.i.d. property, meaning $\tilde{\mathbf{C}}$ may not meet the conditions for exact recovery. Therefore, a crucial part of our analysis is the careful construction of the measurement matrices $\mathbf{A}$ and $\mathbf{B}$ to ensure that the resulting sparse recovery problem has a measurement matrix with the desired structure for unique recovery, as detailed below.

**Construction of measurement matrices:** First, note that $\mathrm{Col}(\mathbf{A})^\perp$ is an $r$-dimensional subspace of $\mathbb{R}^m$, where $r = \dim \mathrm{Col}(\mathbf{A})^\perp$. Given $\mathbf{A} \in \mathbb{R}^{m \times p}$, either generated from a deterministic or random model, we first form the matrix $\mathbf{H} = \mathbf{AG}^\dagger$. We then sample $r$ i.i.d. random vectors $\mathbf{c}_1, \mathbf{c}_2, \ldots, \mathbf{c}_r$ from some distribution $F$ on $\mathbb{R}^p$, meaning each sampled vector is a random vector of size $\mathbb{R}^p$. Let $\mathbf{C} = \frac{1}{\sqrt{r}}\sum_{i=1}^r \mathbf{e}_i\mathbf{c}_i^T$ be the scaled measurement matrix. Before we proceed to construct the matrix $\tilde{\mathbf{C}}$, we make the following observations. Let the size of $\mathbf{G}$ be $q \times p$. Since $\mathbf{G}$ has full column rank, the rank of $\mathbf{G}$ is $p$. It then follows that the rank of $\mathbf{G}^\dagger$ is also $p$. Considering the matrix $\mathbf{H} = \mathbf{AG}^\dagger$, since $\mathbf{G}^\dagger$ has full row rank, the rank of $\mathbf{H} \in \mathbb{R}^{m \times q}$ is the rank of $\mathbf{A}$. Hence, $\dim \mathrm{Col}(\mathbf{H})^\perp = r$. We form a matrix $\tilde{\mathbf{C}}$ of size $m \times p$ where its $r$ rows are the sampled random vectors, scaled by $\frac{1}{\sqrt{r}}$, and the remaining $m - r$ rows are such that each column of $\tilde{\mathbf{C}}$ lies in $\mathrm{Col}(\mathbf{H})^\perp$. Note this can always be achieved since $\mathrm{Col}(\mathbf{H})^\perp$ is an $r$-dimensional space in $\mathbb{R}^m$. Therefore, any point in $\mathrm{Col}(\mathbf{H})^\perp$ can be realized by picking the first $r$ points at random and determining the rest of the entries from the constraints that define $\mathrm{Col}(\mathbf{H})^\perp$. It is important to note that the rows of $\tilde{\mathbf{C}}$ are not i.i.d. (except the first $r$ rows generated from the random model). The matrix $\mathbf{C}$ will be essential to our main analysis, as its rows are sampled from the distribution $F$. We now form the matrix $\mathbf{B} \in \mathbb{R}^{m \times n}$ by constructing each of its columns as follows: $\mathbf{b}_i = \tilde{\mathbf{c}}_i + \mathbf{h}_i$, where $\mathbf{b}_i$ is the $i$-th column of $\mathbf{B}$, $\tilde{\mathbf{c}}_i$ is the $i$-th column of $\tilde{\mathbf{C}}$, and $\mathbf{h}_i$ is an arbitrary vector that is in the column space of $\mathbf{H}$. This ensures that $\tilde{\mathbf{C}} = \mathcal{P}_{\mathrm{Col}(\mathbf{A})}$ as desired. Figure 2 summarizes the construction of the measurement matrices.

**Theorem 6.** *Let $F$ a distribution of random vectors on $\mathbb{R}^p$ and $\mathbf{c}_1, \mathbf{c}_2, \ldots, \mathbf{c}_r$ be $r$ vectors sampled from $F$. The constants $\mu(F)$ and $\kappa_s$ denote the coherence and sparse condition number associated to $F$. Let $\mathbf{z}^* = \mathbf{G}^\dagger\mathbf{x}^* \in \mathbf{T}$, where $\mathbf{T} = \mathrm{Ker}(\mathcal{P}_{\mathrm{Col}(\mathbf{B})^\perp}\mathbf{H})^\perp$ and $\mathbf{u}^*$ be an $s$-sparse vector. Set $\mathbf{y} = \mathbf{Ax}^* + \mathbf{Bu}^*$. Let $\beta \geq 1$ and $r = \dim\mathrm{Col}(\mathbf{A})^\perp$. If $r \geq C\kappa_s\,\mu(F)\,\beta^2\,s\log n$, $(\mathbf{x}^*, \mathbf{u}^*)$ is the unique minimizer to (9) with probability at least $1 - e^{-\beta}$.*

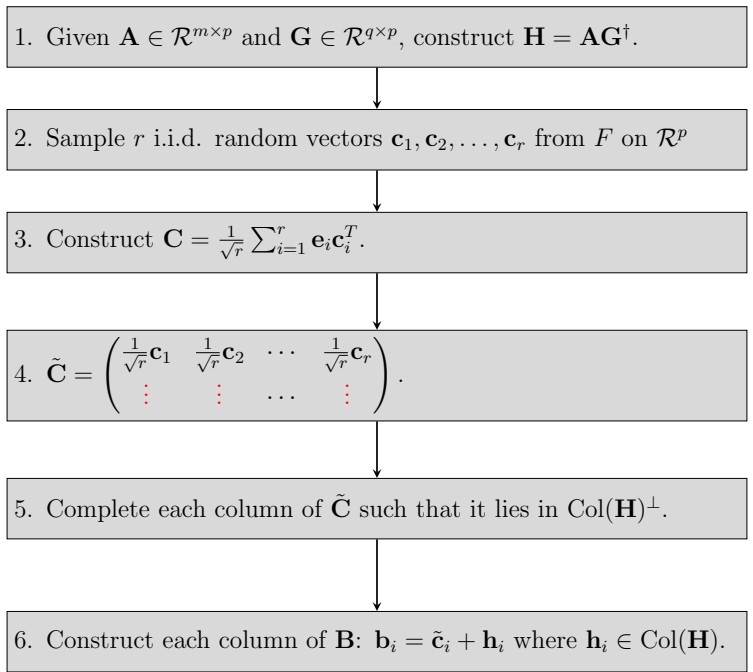

Figure 2: Flow chart that shows the construction of the measurement matrices.

*Proof.* We make a change of variable $\mathbf{z} = \mathbf{Gx}$ and rewrite the optimization in (9) as follows

$$\min_{\mathbf{z},\mathbf{u}} \ ||\mathbf{z}||_2^2 + ||\mathbf{u}||_1 \quad \text{subject to} \quad \mathbf{y} = \mathbf{AG}^\dagger\mathbf{z} + \mathbf{Bu} = \mathbf{Hz} + \mathbf{Bu}. \tag{10}$$

We structure the proof into 3 parts.

**1. Optimality in the mixed objective**

We consider a generic feasible solution pair $(\mathbf{z}^* + \boldsymbol{\delta_1}, \mathbf{u}^* + \boldsymbol{\delta_2})$. Let the function $f(\mathbf{z}, \mathbf{u})$ define the objective in the optimization program. The idea of the proof is to show that any feasible solution is not minimial in the objective value, $f(\mathbf{z}^* + \boldsymbol{\delta_1}, \mathbf{u}^* + \boldsymbol{\delta_2}) \geq f(\mathbf{z}^*, \mathbf{u}^*)$, with the inequality holding for all choices of $\boldsymbol{\delta_1}$ and $\boldsymbol{\delta_2}$ and equality obtained if and only if $\boldsymbol{\delta_1} = \boldsymbol{\delta_2} = \mathbf{0}$. Before we proceed, two remarks are in order. First, using the duality of the $\ell_1$ norm and the $\ell_\infty$ norm, there exists a $\boldsymbol{\Lambda}$ with $\text{supp}(\boldsymbol{\Lambda}) \subset S^c$, $||\boldsymbol{\Lambda}||_\infty = 1$ and such that $\langle \boldsymbol{\Lambda}, (\boldsymbol{\delta_2})_{S^c} \rangle = ||(\boldsymbol{\delta_2})_{S^c}||_1$. Second, the subgradient of the $\ell_1$ norm at $\mathbf{u}^*$ is characterized as follows: $\partial||\mathbf{u}^*||_1 = \{\text{sgn}(\mathbf{u}_S^*) + \mathbf{g} \mid \text{supp}(\mathbf{g}) \in S^c, ||\mathbf{g}_{S^c}||_\infty \leq 1\}$. It follows that $\text{sgn}(\mathbf{u}^*) + \boldsymbol{\Lambda}$ is a subgradient of the $\ell_1$ norm at $\mathbf{u}^*$. It then follows that the inequality $||\mathbf{u}||_1 \geq ||\mathbf{u}^*||_1 + \langle \text{sgn}(\mathbf{u}_S^*) + \boldsymbol{\Lambda}, \mathbf{u} - \mathbf{u}^* \rangle$ holds for any $\mathbf{u}$. We lower bound $f(\mathbf{z}^* + \boldsymbol{\delta_1}, \mathbf{u}^* + \boldsymbol{\delta_2})$ as follows.

$$\begin{aligned} ||\mathbf{z}^* + \boldsymbol{\delta_1}||_2^2 + ||\mathbf{u}^* + \boldsymbol{\delta_2}||_1 \geq & ||\mathbf{z}^*||_2^2 + ||\boldsymbol{\delta_1}||^2 + 2\langle \mathbf{z}^*, \boldsymbol{\delta_1} \rangle + ||\mathbf{u}^*||_1 + \langle \text{sgn}(\mathbf{u}_S^*) + \boldsymbol{\Lambda}, \boldsymbol{\delta_2} \rangle \\ = & f(\mathbf{z}^*, \mathbf{u}^*) + ||\boldsymbol{\delta_1}||^2 + \langle \text{sgn}(\mathbf{u}_S^*) + \boldsymbol{\Lambda}, \boldsymbol{\delta_2} \rangle. \end{aligned} \tag{11}$$

Above, the second equality uses the fact that $\mathbf{z}^* \in \mathbf{T}$ and $\boldsymbol{\delta_1} \in \mathbf{T}^\perp$. The latter fact follows from the feasibility condition that $\mathbf{H}\boldsymbol{\delta_1} + \mathbf{B}\boldsymbol{\delta_2} = \mathbf{0}$, and the application of the projection operator $\mathcal{P}_{\text{Col}(\mathbf{B})^\perp}$ on both sides.

**2. Introducing the dual certificate v**

We next use the feasibility condition that $\mathbf{H}\boldsymbol{\delta_1} + \mathbf{B}\boldsymbol{\delta_2} = \mathbf{0}$ and introduce the dual certificate $\mathbf{v}$. To eliminate the component $\mathbf{H}\boldsymbol{\delta_1}$, we project it onto the orthogonal complement of the range of $\mathbf{H}$ and obtain $\tilde{\mathbf{C}}\boldsymbol{\delta_2} = \mathbf{0}$ where $\tilde{\mathbf{C}} = \mathcal{P}_{\text{Col}(\mathbf{H})^\perp}(\mathbf{B})$. We note that the previous relation follows from the construction of the measurement matrices. We further consider a reduced system by considering $\mathbf{C}\boldsymbol{\delta_2} = \mathbf{0}$. Recall that $\mathbf{C}$ is a matrix where each row is sampled i.i.d. from the distribution $F$. With that, we essentially have a sparse recovery problem with the measurement matrix $\mathbf{C}$. Using Lemma 1, and the assumptions of the theorem, there exists a dual

certificate $\mathbf{v} \in \text{Col}(\mathbf{C}^T)$. With this, we continue with lower bounding $f(\mathbf{z}^* + \boldsymbol{\delta_1}, \mathbf{u}^* + \boldsymbol{\delta_2})$.

$$||\mathbf{z}^* + \boldsymbol{\delta_1}||_2^2 + ||\mathbf{u}^* + \boldsymbol{\delta_2}||_1 \geq ||\mathbf{z}^*||_2^2 + ||\boldsymbol{\delta_1}||_2^2 + ||\mathbf{u}^*||_1 + \langle \text{sgn}(\mathbf{u}_S^*) + \boldsymbol{\Lambda} - \mathbf{v}, \boldsymbol{\delta_2} \rangle$$
$$= f(\mathbf{x}^*, \mathbf{u}^*) + \langle \text{sgn}(\mathbf{u}_S^*) + \boldsymbol{\Lambda} - \mathbf{v}, \boldsymbol{\delta_2} \rangle. \tag{12}$$

It remains to show that $\langle \text{sgn}(\mathbf{u}_S^*) + \boldsymbol{\Lambda} - \mathbf{v}, \boldsymbol{\delta_2} \rangle > 0$. By considering projections onto $S$ and $S^c$ and using the fact that $\langle \boldsymbol{\Lambda}, (\boldsymbol{\delta_2})_{S^c} \rangle = ||(\boldsymbol{\delta_2})_{S^c}||_1$, we obtain

$$\langle \text{sgn}(\mathbf{u}_S^*) + \boldsymbol{\Lambda} - \mathbf{v}, \boldsymbol{\delta_2} \rangle = \langle \text{sgn}(\mathbf{u}_S^*) - \mathbf{v}_S, (\boldsymbol{\delta_2})_S \rangle - \langle \mathbf{v}_{S^c}, (\boldsymbol{\delta_2})_{S^c} \rangle + ||(\boldsymbol{\delta_2})_{S^c}||_1$$
$$\geq -||\text{sgn}(\mathbf{u}_S^*) - \mathbf{v}_S||_2 ||(\boldsymbol{\delta_2})_S||_2 - ||\mathbf{v}_{S^c}||_\infty ||(\boldsymbol{\delta_2})_{S^c}||_1 + ||(\boldsymbol{\delta_2})_{S^c}||_1$$
$$\geq -\frac{1}{4}||(\boldsymbol{\delta_2})_S||_2 - \frac{1}{4}||(\boldsymbol{\delta_2})_{S^c}||_1 + ||(\boldsymbol{\delta_2})_{S^c}||_1 \tag{13}$$
$$= -\frac{1}{4}||(\boldsymbol{\delta_2})_S||_2 + \frac{3}{4}||(\boldsymbol{\delta_2})_{S^c}||_1 \geq \frac{1}{4}||(\boldsymbol{\delta_2})_{S^c}||_1. \tag{14}$$

The inequality in (13) the last inequality follow from the conditions on $\mathbf{v}$ from Lemma 1. Combining (12) and the above bound with the final result noted in (14), we have

$$f(\mathbf{z}^* + \boldsymbol{\delta_1}, \mathbf{u}^* + \boldsymbol{\delta_2}) \geq f(\mathbf{z}^*, \mathbf{u}^*) + ||\boldsymbol{\delta_1}||_2^2 + \frac{1}{4}||(\boldsymbol{\delta_2})_{S^c}||_1. \tag{15}$$

**3. Uniqueness of solution**
We note that $f(\mathbf{z}^* + \boldsymbol{\delta_1}, \mathbf{u}^* + \boldsymbol{\delta_2}) = f(\mathbf{z}, \mathbf{u})$ if and only if $||(\boldsymbol{\delta_2})_{S^c}||_1 = 0$ and $\boldsymbol{\delta_1} = 0$. Since $||(\boldsymbol{\delta_2})_S||_2 \leq 2||(\boldsymbol{\delta_2})_{S^c}||_2$, the equality $||(\boldsymbol{\delta_2})_{S^c}||_1 = 0$ implies that $||(\boldsymbol{\delta_2})_S||_2 = 0$. With this, $f(\mathbf{x}^* + \boldsymbol{\delta_1}, \mathbf{u}^* + \boldsymbol{\delta_2}) = f(\mathbf{x}, \mathbf{u})$ if and only if $\boldsymbol{\delta_2} = \mathbf{0}$. Therefore, the solution $(\mathbf{z}^*, \mathbf{u}^*)$ achieves the minimal value in the objective, and is a unique solution to (10). Finally, using the relation $\mathbf{x}^* = G^\dagger \mathbf{z}^*$ guarantees unique solution to (9). This concludes the proof. $\qquad \square$

**Remark 1.** *Consider the following optimization program*

$$\min_{\mathbf{x}, \mathbf{u}} ||\mathbf{G}\mathbf{x}||_2^2 + \lambda ||\mathbf{u}||_1 \quad subject\ to \quad \mathbf{y} = \mathbf{A}\mathbf{x} + \mathbf{B}\mathbf{u}, \tag{16}$$

*where $\lambda > 0$ is a constant that balances the two terms in the objective. Following the same steps as in the proof of Theorem 6 also guarantees unique solutions for the modified objective.*

## 4.3 Special cases of analysis and discussion

We note a few special cases of our main theorem in Theorem 6. First, when the Tikhonov matrix $\mathbf{G}$ is the identity matrix, the regularization on $\mathbf{x}$ is the standard $\ell_2$ (minimum norm least squares) regularization. The main result continues to hold, with additional assumptions, when $\mathbf{G} = \mathbf{A}$. In this case, the optimization problem is

$$\min_{\mathbf{x} \in \text{Ker}(\mathbf{A})^\perp, \mathbf{u}} ||\mathbf{A}\mathbf{x}||_2^2 + ||\mathbf{u}||_1 \quad subject\ to \quad \mathbf{y} = \mathbf{A}\mathbf{x} + \mathbf{B}\mathbf{u}. \tag{17}$$

We note that the uniqueness result for the above problem is modulo restriction to the subspace $\text{Ker}(\mathbf{A})^\perp$. We now state the following result.

**Theorem 7.** *Let $F$ a distribution of random vectors on $\mathbb{R}^p$ and $\mathbf{c}_1, \mathbf{c}_2, \ldots, \mathbf{c}_r$ be $r$ vectors sampled from $F$. The constants $\mu(F)$ and $\kappa_s$ denote the coherence and sparse condition number associated to $F$. Set $\mathbf{y} = \mathbf{A}\mathbf{x}^* + \mathbf{B}\mathbf{u}^*$. Let $\beta \geq 1$ and $r = dim\,\text{Col}(\mathbf{A})^\perp$. Assume the two conditions*

$$||\mathbf{B}_S^T \mathbf{A}|| \leq \frac{1}{32||\mathbf{x}^*||_2}, \quad \max_{i,j} |(\mathbf{B}_{S^c}^T \mathbf{A})_{i,j}| \leq \frac{1}{32||\mathbf{x}^*||_\infty}. \tag{18}$$

*If $r \geq C \kappa_s \mu(F) \beta^2 s \log n$, $(\mathbf{x}^*, \mathbf{u}^*)$ is the unique minimizer to (17) with probability at least $1 - e^{-\beta}$.*

*Proof.* The proof follows the proof of Theorem 6. The main difference is that we need to show that $\langle \text{sgn}(\mathbf{u}_S^*) + \mathbf{\Lambda} - \mathbf{v} - 2\mathbf{B}^T\mathbf{A}\mathbf{x}^*, \boldsymbol{\delta_2}\rangle > 0$. By considering projections onto $S$ and $S^c$ and using the fact that $\langle \mathbf{\Lambda}, (\boldsymbol{\delta_2})_{S^c}\rangle = ||(\boldsymbol{\delta_2})_{S^c}||_1$, we obtain

$$\langle \text{sgn}(\mathbf{u}_S^*) + \mathbf{\Lambda} - \mathbf{v} - 2\mathbf{B}^T\mathbf{A}\mathbf{x}^*, \boldsymbol{\delta_2}\rangle$$
$$= \langle \text{sgn}(\mathbf{u}_S^*) - \mathbf{v}_S, (\boldsymbol{\delta_2})_S\rangle - \langle \mathbf{v}_{S^c}, (\boldsymbol{\delta_2})_{S^c}\rangle + ||(\boldsymbol{\delta_2})_{S^c}||_1 - \langle 2[\mathbf{B}^T\mathbf{A}\mathbf{x}^*]_S, (\boldsymbol{\delta_2})_S\rangle$$
$$- \langle 2[\mathbf{B}^T\mathbf{A}\mathbf{x}^*]_{S^c}, (\boldsymbol{\delta_2})_{S^c}\rangle$$
$$\geq -||\text{sgn}(\mathbf{u}_S^*) - \mathbf{v}_S||_2 ||(\boldsymbol{\delta_2})_S||_2 - ||\mathbf{v}_{S^c}||_\infty ||(\boldsymbol{\delta_2})_{S^c}||_1 + ||(\boldsymbol{\delta_2})_{S^c}||_1$$
$$- 2||\mathbf{B}_S^T\mathbf{A}|| \, ||\mathbf{x}^*||_2 ||(\boldsymbol{\delta_2})_S||_2 - 2 \max_{i,j}|(\mathbf{B}_{S^c}^T\mathbf{A})_{i,j}| ||\mathbf{x}^*||_\infty ||(\boldsymbol{\delta_2})_{S^c}||_1$$
$$\geq -\frac{1}{4}||(\boldsymbol{\delta_2})_S||_2 - \frac{1}{4}||(\boldsymbol{\delta_2})_{S^c}||_1 + ||(\boldsymbol{\delta_2})_{S^c}||_1 - \frac{1}{16}||(\boldsymbol{\delta_2})_S||_2 - \frac{1}{16}||(\boldsymbol{\delta_2})_{S^c}||_1$$
$$= -\frac{5}{16}||(\boldsymbol{\delta_2})_S||_2 + \frac{11}{16}||(\boldsymbol{\delta_2})_{S^c}||_1$$
$$\geq -\frac{10}{16}||(\boldsymbol{\delta_2})_{S^c}||_1 + \frac{11}{16}||(\boldsymbol{\delta_2})_{S^c}||_1 = \frac{1}{16}||(\boldsymbol{\delta_2})_{S^c}||_1.$$

The rest of the proof is similar to the proof of Theorem 6. □

## 4.4 Comparison to compressive sensing

We note that the dense and sparse coding problem problem can equivalently be formulated as $\mathbf{y} = \mathbf{A}\mathbf{x} + \mathbf{B}\mathbf{u} = [\mathbf{A} \ \mathbf{B}] \begin{bmatrix} \mathbf{x} \\ \mathbf{u} \end{bmatrix}$. Applying standard compressive sensing (CS) results imposes uniform conditions on $[\mathbf{A} \ \mathbf{B}]$, which may lead to sub-optimal recovery guarantees. We will use Theorem 5 to illustrate the difference between the CS approach and ours. For a concrete example, let $\mathbf{A}$ be a $20 \times 20$ invertible matrix, $\mathbf{B}$ be $20 \times 80$ matrix and set $k = 2$. To recover $\begin{bmatrix} \mathbf{x} \\ \mathbf{u} \end{bmatrix}$ exactly, one condition based on CS (see Theorem 1.7 in Davenport et al. (2012)) is:

$$p + k < \frac{1}{2}\left(1 + \frac{1}{\mu[\mathbf{A} \ \mathbf{B}]}\right) \rightarrow k < \frac{1}{2}\left(1 + \frac{1}{\mu[\mathbf{A} \ \mathbf{B}]}\right) - p,$$

where $\mu[\mathbf{A} \ \mathbf{B}]$ denotes the mutual coherence of the combined dictionary. Note that the range of $\mu[\mathbf{A} \ \mathbf{B}]$ is $[\mu_0, 1]$ where $\mu_0$ denotes the Welch bound (see Definition 1.5 in Davenport et al. (2012)). We make the following observations:

- The maximum sparsity decreases as $p$ increases. With $p = 20$, mutual coherence of the combined dictionary needs to be at most $1/43 \approx 0.0233$.

- In our approach, to allow for sparsity $k = 2$, the block coherence needs to be at most $1/(2\sqrt{20}) \approx 0.1118$.

- Theorem 5 relies on block coherence and does not require coherence within the blocks $\mathbf{A}$ and $\mathbf{B}$. This provides a flexible model, allowing $\mathbf{A}$ to be a deterministic coherent dictionary, for example.

Next, consider applying $\ell_1$ minimization and standard compressive sensing theory to guarantee uniqueness given the measurement matrix $\mathbf{R} = [\mathbf{A} \ \mathbf{B}]$. One guarantee based on incoherence (see Theorem 1.1 in Candes & Plan (2011)) states that the number of measurements must be on the order of $\mu_*(p + k)\log(p + n)$, where $\mu_*$ is the coherence of the combined dictionary $\mathbf{R}$. In contrast to the mutual coherence mentioned earlier, $\mu_*$ is the smallest number such that the following equality holds for any row of $\mathbf{R}$ denoted by $\mathbf{a}$:

$$\max_{1 \leq t \leq (p+n)} |a(i)|^2 < \mu_*,$$

where $a(i)$ denotes the i-th entry of $\mathbf{a}$. It is typically assumed that there is an underlying distribution $F$ from which the rows of $\mathbf{R} = [\mathbf{A} \ \mathbf{B}]$ are sampled independently and identically. We note that the range of $\mu_*$, for the measurement setup in Candes & Plan (2011), is $[1, n + p]$. The implication of this is that the sample complexity implicitly requires $\mu_* = O(1)$. In the case of mixed dictionaries, as in our setup, a coherent matrix $\mathbf{A}$ can lead to sub-optimal number of measurements.

## 5 Experiments

The codes for reproducing the experiments in this section can be found on GitHub: `https://github.com/manosth/densae/` and `https://github.com/btolooshams/densae`.

### 5.1 Phase transition curves

We generate phase transition curves and present how the success rate of the recovery, using the proposed model, changes under different scenarios. To generate the data, we sample random matrices $\mathbf{A} \in \mathbb{R}^{m \times p}$ and $\mathbf{B} \in \mathbb{R}^{m \times n}$ whose columns have expected unit norm and fix the number of columns of $\mathbf{B}$ to be $n = 100$. The vector $\mathbf{u} \in \mathbb{R}^n$ has $s$ randomly chosen indices, whose entries are drawn according to a standard normal distribution, and $x \in \mathbb{R}^p$ is generated as $\mathbf{x} = \mathbf{A}^T \gamma$ where $\gamma \in \mathbb{R}^m$ is a random vector. The construction ensures that $\mathbf{x}$ does not belong in the null space of $\mathbf{A}$, and hence degenerate cases are avoided. We normalize both $\mathbf{x}$ and $\mathbf{u}$ to have unit norm, and generate the measurement vector $\mathbf{y} \in \mathbb{R}^m$ as $\mathbf{y} = \mathbf{A}\mathbf{x} + \mathbf{B}\mathbf{u}$.

To generate the transition curves we vary the *sampling ratio* $\sigma = \frac{m}{n+p} \in [0.05, 0.95]$ and the *sparsity ratio* $\rho = \frac{s}{m}$ in the same range. Note that the sensing matrix in our model is $[\mathbf{A} \ \mathbf{B}]$; therefore, our definition of $\sigma$ takes into account both the size of $\mathbf{A}$ and $\mathbf{B}$. In the case where we revert to the compressive sensing scenario ($p = 0$), the sampling ratios coincide. We solve the convex optimization problem of (17) to obtain the numerical solution pair $(\hat{\mathbf{x}}, \hat{\mathbf{u}})$ using `CVXPY` (Diamond & Boyd, 2016; Agrawal et al., 2018), and register a successful recovery if both $\frac{\|\hat{\mathbf{x}} - \mathbf{x}\|_2}{\|\mathbf{x}\|_2} \leq \epsilon$ and $\frac{\|\hat{\mathbf{u}} - \mathbf{u}\|_2}{\|\mathbf{u}\|_2} \leq \epsilon$, with $\epsilon = 10^{-3}$. For each choice of $\sigma$ and $\rho$, we average 100 independent runs to estimate the success rate.

Figure 3 shows the phase transition curves, indicating the probability of successful recovery, for $p \in \{0.1m, 0.5m\}$ to highlight different ratios between $p$ and $n$. We observe that increasing $p$ leads to a deterioration in performance. This is expected, as this creates a greater overlap on the spaces spanned by $\mathbf{A}$ and $\mathbf{B}$. We can view our formulation as modeling the noise of the system. In such a case, the number of columns of $\mathbf{A}$ encodes the complexity of the noise system: as $p$ increases, so does the span of the noise space. Extending the signal processing interpretation, note that we model the noise signal $\mathbf{x}$ as a dense vector, which can be seen as encoding smooth areas of the signal that correspond to *low-frequency* components. On the contrary, the signal $\mathbf{u}$ has, by construction, a sparse structure, containing *high-frequency* information, an interpretation that will be further validated on real data in Section 5.4.

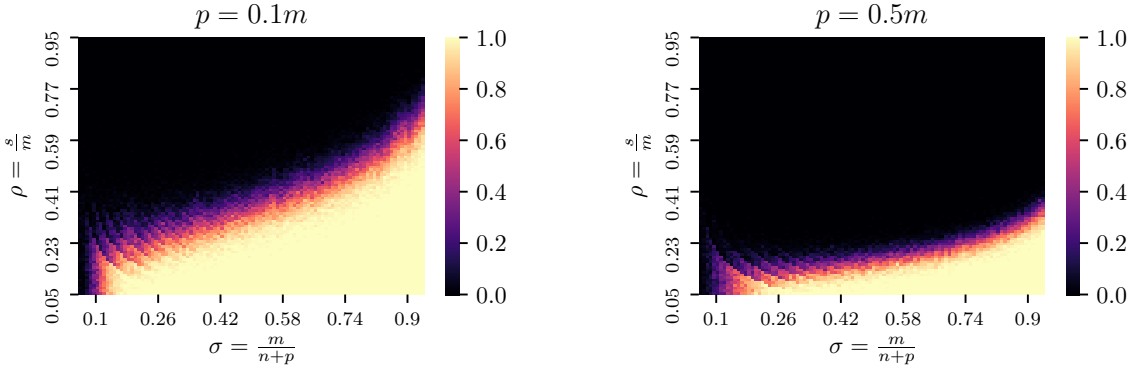

Figure 3: **Phase transition curves for $p = 0.1m$ (left) and $p = 0.5m$ (right)**. Colors represent the probability of successful recovery, ranging from black (vector recovery failed in all trials) to yellow (recovery was always successful).

### 5.2 Noisy compressive sensing

If $\mathbf{x}$ can be interpreted as a noise vector, how does our model compare to noisy compressive sensing? Compressive sensing can be extended to the noisy case, which allows for the successful recovery of sparse

signals under the presence of noise, assuming an upper bound on the noise level. In the rest of the section we examine how the proposed model, which incorporates both sparse and dense components, fares against this noisy variant.

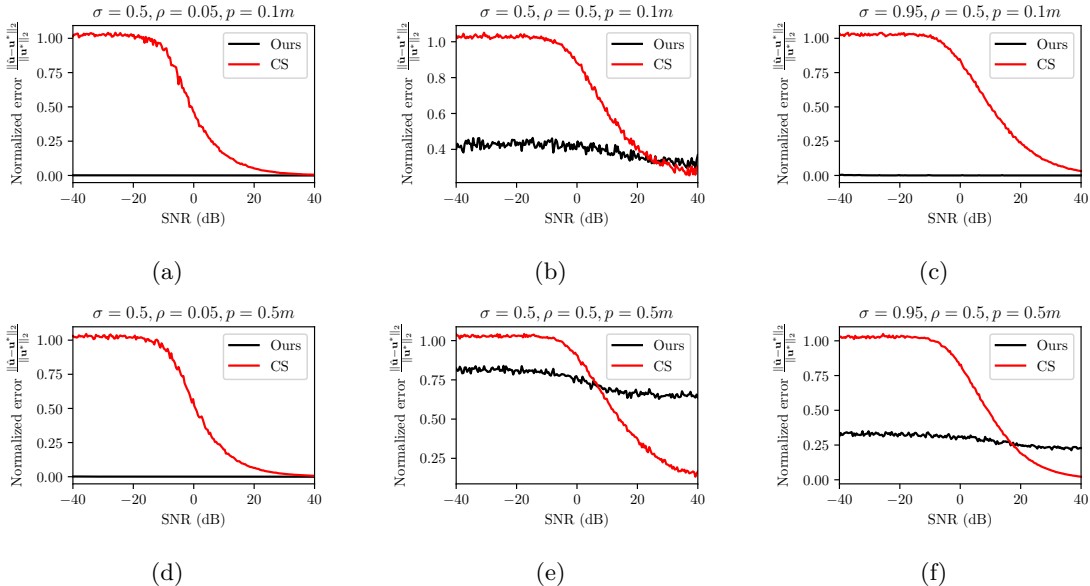

Figure 4: Normalized recovery error of $\boldsymbol{u}$ as the SNR varies (lower is better).

We, again, fix the number of columns in $\boldsymbol{B}$ to be $n = 100$, and generate the matrices $\boldsymbol{A} \in \mathbb{R}^{m \times p}$ and $\boldsymbol{B} \in \mathbb{R}^{m \times n}$, as well as the vectors $\boldsymbol{x}^* \in \mathbb{R}^p$ and $\mathbf{u}^* \in \mathbb{R}^n$, as before. We define the signal-to-noise ratio as $\text{SNR} = 20 \log_{10} \frac{||\boldsymbol{u}^*||_2}{||\boldsymbol{x}^*||_2}$, and iterate over the range $[-40\text{dB}, 40\text{dB}]$. To vary the SNR, we normalize both vectors and scale $\mathbf{u}^*$ by $10^{\frac{\text{SNR}}{20}}$. For our proposed method, we solve the optimization of (9), whereas for noisy compressive sensing we solve

$$\hat{\mathbf{u}} = \arg\min_{\mathbf{u}} \ ||\mathbf{u}||_1 \quad \text{subject to} \quad ||\mathbf{y} - \mathbf{B}\mathbf{u}||_2 \leq ||\mathbf{A}\mathbf{x}^*||_2, \tag{19}$$

and report the normalized error $\frac{||\hat{\mathbf{u}} - \mathbf{u}^*||_2}{||\mathbf{u}^*||_2}$ for the two methods averaging 100 independent runs.

We present the SNR curves in Figure 4. As an initial observation, note that noisy compressive sensing, in every case, recovers a vector when the energy of the noise is less than that of the signal (for a SNR> 0dB), and in most scenarios full recovery is not achieved unless the ratio is very large (SNR> 25dB). That is expected, since the results for those settings assume an upper bound on the noise level. In contrast, our proposed model is able to recover both signals even when the norm of $\mathbf{x}$ is 100 times larger than that of $\mathbf{u}$, at the cost of having access to the additional measurement matrix $\mathbf{A}$.

For low sparsity ratios (Figures 4(a) and (d)), we are able to recover both signals in every experiment using our model, whereas compressive sensing exhibits the behaviour we discussed above. Increasing the sparsity ratio while keeping the number of samples the same (Figures 4(b) and (e)) results in a significant reduction in performance for both models (note the different axis scaling for these figures). This is in line with both the results for noisy compressive sensing and our results presented in Section 4 (as well as the phase transition curves of the previous subsection). When the overlap between $\mathbf{A}$ and $\mathbf{B}$ is greater (Figure 4(e)) our model suffers a greater performance hit, as is expected from our analysis; note that noisy compressive sensing is unaffected by the relative size of $\mathbf{A}$ and $\mathbf{B}$, since it only makes assumptions about the noise level and not its span. Finally, we observe that increasing the number of measurements (Figures 4(c) and (f)) restores performance, in line with our analysis.

**Remark:** Note that the large sparsity ratios ($\rho = 0.5$, corresponding to Figures 4(b), (c), (e), and (f)) violate our assumptions in Section 4 and recovery is not expected based on the transition curves of the previous

subsection. However, we include them for illustration, as for smaller values of $\rho$ our model always recovered both vectors. As a key takeaway from this line of experimentation, we recommend the use of our model in every case where there is a structured interference with significant norm. In cases where the norm of the signal of interest is dominating (SNR$\approx$ 40dB) and there is some form of degeneracy (there is a large overlap between the signal spaces and $\mathbf{u}$ is barely sparse), noisy compressive sensing may be more appropriate.

### 5.3 Sensing with real data

In this section, we present experiments on real data using random sensing matrices. The real data we consider is the MNIST database of handwritten digits (LeCun, 1998; LeCun et al., 1998). Through these experiments, we want to (i) showcase the performance of our model on actual, real data and (ii) show the efficacy of our algorithm when using overcomplete sensing matrices. To generate overcomplete dictionaries, we proceeded as

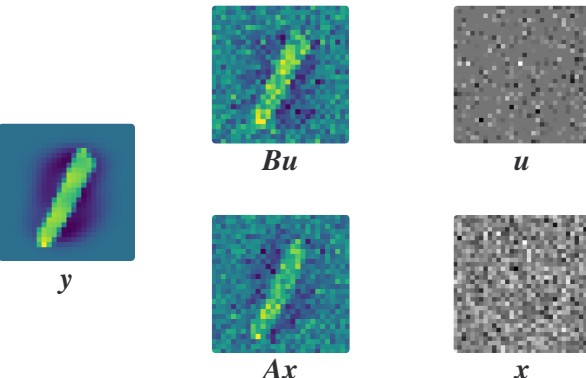

Figure 5: **Decomposition of an MNIST image to its sparse and dense components**. We visualize the minimization of the terms in (20): the input $\mathbf{y}$ is adequately reconstructed (left), the component $\mathbf{Ax}$ is smooth relative to $\mathbf{Bu}$ (middle), and $\mathbf{u}$ is sparse.

follows: as MNIST images can be seen as vectors $\boldsymbol{y} \in \mathbb{R}^{784}$, we first generated a random orthogonal matrix in $\mathbb{R}^{784 \times 784}$. We used half of the columns of that matrix as a base for $\mathbf{A} \in \mathbb{R}^{784 \times 1024}$ and the other half for $\mathbf{B} \in \mathbb{R}^{784 \times 1024}$; the rest of the columns were generated as linear combinations of each base. To avoid artificial orderings that might result when using certain optimizers, we conclude the matrix generation with random column permutations. These matrices, while in combination they do span $\mathbb{R}^{784}$, were not the generating model for the MNIST dataset. As such, we slightly alter the optimization problem of (9) to relax the exact reconstruction, yielding:

$$\min_{\mathbf{x}, \mathbf{u}} ||\mathbf{Ax} + \mathbf{Bu} - \mathbf{y}||_2^2 + \mu ||\mathbf{Ax}||_2^2 + \lambda ||\mathbf{u}||_1 . \tag{20}$$

In our experiments, we use $\mu = 3$ and $\lambda = 0.01$. A visual decomposition of such an image is presented in Figure 5.

Both $\mathbf{u}$ and $\mathbf{x}$ have their expected properties: $\mathbf{u}$ is visibly sparse, with few nonzero elements and $\mathbf{x}$ has a fairly dense structure. Moreover, we observe that the component of $\boldsymbol{Ax}$ seems slightly more fainted compared to that of $\boldsymbol{Bu}$.

To further validate this observation we computed both the Euclidean norm and total variation for each component, as a proxy for smoothness, and plotted the distributions of total variation in Figure 6. The distributions were computed using 10000 training samples. The distribution of the Euclidean norm was very similar and can be found in the Appendix, Figure 13. Observing the distributions of $\boldsymbol{Ax}$ and $\boldsymbol{Bu}$ we note that the distribution corresponding to the sparse component is skewed to higher values of total variation. This empirically validates the effectiveness of the smoothness regularization of (9), even when sensing using random matrices. As a final remark, we attempted to use $\mathbf{B}$ in a sparse coding framework to recover a sparse vector. However, in every instance the solver *failed* to converge and produce a sparse vector; this is, to an

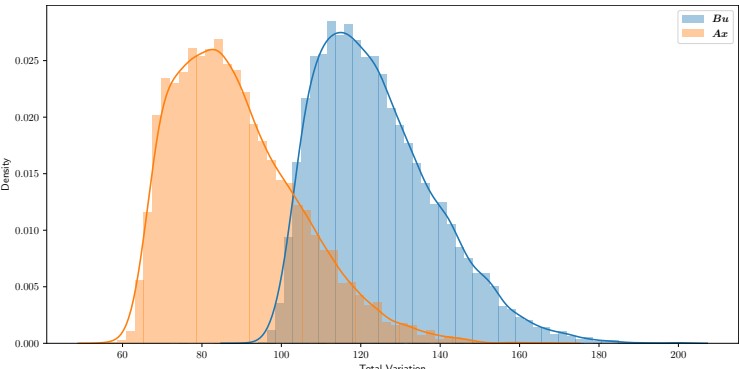

Figure 6: **Total variation distribution for the components $Ax$ and $Bu$ of MNIST images**. We quantify the qualitative difference in smoothness between $\mathbf{Ax}$ and $\mathbf{Bu}$ in (20), supporting the qualitative difference of Figure 5.

extent, expected as $\mathbf{B}$ was generated using only half the columns of an orthogonal matrix and therefore is unable to fully span the image space, failing to adequately represent all the images in the data set.

The main analysis in this paper is based on the measurement matrices and the Tikhonov matrix satisfying certain conditions, such as randomness in the matrix $\mathbf{B}$. However, in practice, efficient matrices are not always available, and random matrices might not be optimal for every data set. Dictionary learning refers to the problem of learning $\mathbf{A}$ and $\mathbf{B}$ from data (Agarwal et al., 2014; Chatterji & Bartlett, 2017; Garcia-Cardona & Wohlberg, 2018). Based on unrolled neural architectures for the sparse coding model ($\mathbf{B} = \mathbf{0}$) (Tolooshams et al., 2020; Gregor & Lecun, 2010), the next section proposes an unrolled autoencoder to infer dense and sparse representations and learn dictionaries from data.

### 5.4 Dictionary learning based neural architecture for dense and sparse coding

We mitigate the drawbacks of convolutional sparse coding model in capturing a wide range of features from natural images by proposing dense and sparse dictionary learning; the framework learns a diverse set of features (smooth features via $\mathbf{A}$ and high-frequency features appearing sparsely through $\mathbf{B}$). We formulate the dense and sparse dictionary learning problem as minimizing the objective

$$\min_{\mathbf{A},\mathbf{B},\mathbf{X},\mathbf{U}} \frac{1}{2}\|\mathbf{Y} - \mathbf{AX} - \mathbf{BU}\|_F^2 + \frac{1}{2\lambda_x}\|\mathbf{AX}\|_F^2 + \lambda_u\|\mathbf{U}\|_1, \tag{21}$$

which can be solved using the following alternating minimization steps

$$\mathbf{X}^{(l)}, \mathbf{U}^{(l)} = \arg\min_{\mathbf{X},\mathbf{U}} \frac{1}{2}\|\mathbf{Y} - \mathbf{A}^{(l)}\mathbf{X} - \mathbf{B}^{(l)}\mathbf{U}\|_F^2 + \frac{1}{2\lambda_x}\|\mathbf{A}^{(l)}\mathbf{X}\|_F^2 + \lambda_u\|\mathbf{U}\|_1. \tag{22}$$

$$\mathbf{A}^{(l+1)}, \mathbf{B}^{(l+1)} = \arg\min_{\mathbf{A},\mathbf{B}} \frac{1}{2}\|\mathbf{Y} - \mathbf{AX}^{(l)} - \mathbf{BU}^{(l)}\|_F^2, \tag{23}$$

where $\mathbf{Y} \in \mathbf{R}^{m \times I}$, $\mathbf{X} \in \mathbf{R}^{p \times I}$, and $\mathbf{U} \in \mathbf{R}^{n \times I}$ with $I$ is the total number of data examples. By design choice, we optimize only the reconstruction loss when updating the dictionaries. Based on the above alternating updates, we use deep unrolling to construct an unrolled neural network (Tolooshams et al., 2020; Gregor & Lecun, 2010), which we term the dense and sparse autoencoder (DenSaE), tailored to learning the dictionaries from the dense and sparse model. The encoder maps $\mathbf{Y}$ into a dense matrix $\mathbf{X}_T$ and a sparse one $\mathbf{U}_T$ using two sets of filters of $\mathbf{A}$ and $\mathbf{B}$ through a recurrent network. $\mathbf{A}$ encodes the smooth part of the data (low frequencies), and $\mathbf{B}$ encodes the details of the signal (high frequencies). The encoding is achieved by unrolling $T$ proximal gradient iterations shown below

$$\mathbf{X}_t = \mathbf{X}_{t-1} + \alpha_x\left(\mathbf{A}^{\mathrm{T}}(\mathbf{Y} - \left(1 + \frac{1}{\lambda_x}\right)\mathbf{AX}_{t-1} - \mathbf{BU}_{t-1})\right),$$
$$\mathbf{U}_t = \mathcal{S}_b\left(\mathbf{U}_{t-1} + \alpha_u\mathbf{B}^{\mathrm{T}}(\mathbf{Y} - \mathbf{AX}_{t-1} - \mathbf{BU}_{t-1})\right), \tag{24}$$

where $\alpha_x$ and $\alpha_u$ are step sizes, and $\mathcal{S}_b$, with $b = \alpha_u \lambda_u$, is the activation function; for non-negative sparse coding, the activation is $\text{ReLU}_b(z) = (z - b) \cdot \mathbb{1}_{z \geq b}$, and for general sparse coding, it is $\text{Shrinkage}_b(z) = \text{ReLU}_b(z) - \text{ReLU}_b(-z)$ (Tolooshams et al., 2020).

Having a non-informative prior on $\mathbf{Ax}$ in DenSaE implies that $\lambda_x \to \infty$. The parameters $\alpha_x$, $\alpha_u$, $\lambda_u$ are tuned empirically. The decoder reconstructs the image using the generative model $\mathbf{y} = \mathbf{Ax}_T + \mathbf{Bu}_T$. For classification, we use $\mathbf{u}_T$ and $\mathbf{x}_T$ as inputs to a linear classifier $C$ that maps them to the predicted class $\hat{\mathbf{q}}$. The dictionaries $\mathbf{A}$ and $\mathbf{B}$ are learned via backpropagation. We remark that $b = \alpha_u \lambda_u$. A larger value of $b$ in the proximal mapping $\mathcal{S}_b$ enforces higher sparsity on $\mathbf{u}$, and a smaller value of $\lambda_x$ promotes smoothness on $\mathbf{Ax}$. We learn the dictionaries $\mathbf{A}$ and $\mathbf{B}$, along with the classifier $\mathbf{C}$, by minimizing the weighted reconstruction (Rec.) and classification (Logistic) loss, represented as $(1 - \beta)$ Rec. $+ \beta$ Logistic), where $\beta \in [0, 1]$. We note that, as $\beta$ increases, the network shifts from reconstructing objective into a more predictive one. Figure 7 shows the DenSaE architecture. We examined the following questions

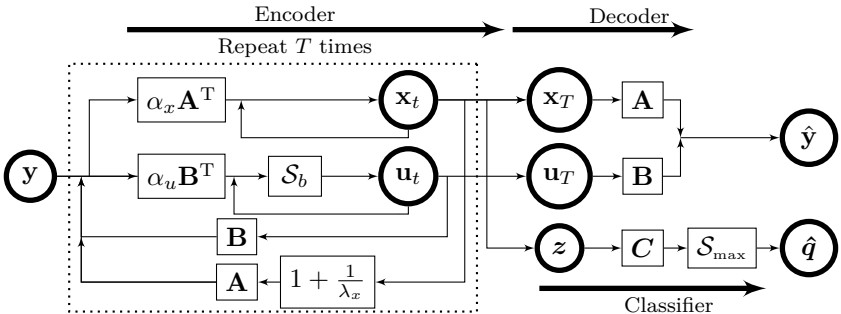

Figure 7: DenSaE. The vector $\mathbf{z}$ comprises the normalized features stacked plus a 1 scalar for the classifier bias, $\mathcal{S}_b$ and $\mathcal{S}_{\max}$ are the soft-thresholding and soft-max operators, respectively.

(i) How do the discriminative, reconstruction, and denoising capabilities change as we vary the number of filters in $\mathbf{A}$ vs. $\mathbf{B}$?

(ii) What is the performance of DenSaE compared to sparse coding networks?

(iii) What data characteristics does the model capture?

As baselines, we trained two variants, $\text{CSCNet}_{\text{hyp}}$ and $\text{CSCNet}_{\text{LS}}$, of CSCNet (Simon & Elad, 2019), an architecture tailored to dictionary learning for the sparse coding problem $\mathbf{y} = \mathbf{Bu}$. Since all three networks are convolutional, $\mathbf{A}$ and $\mathbf{B}$ are Toeplitz matrices that perform the sum of convolutions using multiple filters. In $\text{CSCNet}_{\text{hyp}}$, the bias is tuned as a shared hyper-parameter. In $\text{CSCNet}_{\text{LS}}$, we learn a different bias for each filter by minimizing the reconstruction loss. When the dictionaries are non-convolutional, we call the network SCNet.

### 5.4.1 DenSaE strikes a balance between discriminative capability and reconstruction

We study the case when DenSaE is trained on the MNIST dataset for joint reconstruction and classification. We show (i) how the explicit imposition of sparse and dense representations in DenSaE helps to balance discriminative and representation power, and (ii) that DenSaE outperforms SCNet. We warm start the training of the classifier using dictionaries obtained by first training the autoencoder with $\beta = 0$.

**Characteristics of the representations $\mathbf{x}_T$ and $\mathbf{u}_T$**: To evaluate the discriminative power of the representations learned by only training the autoencoder, we first trained the classifier *given* the representations. Specifically, we first trained $\mathbf{A}$ and $\mathbf{B}$ with $\beta = 0$, then trained $\mathbf{C}$ with $\beta = 1$. We call this disjoint training. Table 1 shows the classification accuracy (Acc.), $\ell_2$ reconstruction loss (Rec.), and the relative contributions, expressed as a percentage, of the dense or sparse representations to classification and reconstruction for disjoint training. Each column of $[\mathbf{A}, \mathbf{B}]$, and of $\mathbf{C}$, corresponds to either a dense or a sparse feature. For

Table 1: DenSaE's performance on MNIST dataset from disjoint training. $\frac{\mathbf{A}}{\mathbf{A+B}}$ is defined as $\frac{\text{\# columns of } \mathbf{A}}{\text{\# columns of } \mathbf{A} + \text{\# columns of } \mathbf{B}} \times 100$.

| | SCNet$_{\text{LS}}$ | SCNet$_{\text{hyp}}$ | $\frac{5\mathbf{A}}{395\mathbf{B}}$ | $\frac{25\mathbf{A}}{375\mathbf{B}}$ | $\frac{200\mathbf{A}}{200\mathbf{B}}$ |
|---|---|---|---|---|---|
| $\frac{\mathbf{A}}{\mathbf{A+B}}$ | - | - | 1.25% | 6.25% | 50% |
| Acc. | 94.16 % | 98.32% | 98.18% | 98.18% | 96.98% |
| Rec. | 1.95 | 6.80 | 6.83 | 6.30 | 3.04 |
| $\mathbf{A}$ contribution to important class features | - | - | 0% | 0% | 0% |
| $\mathbf{A}$ contribution to important rec. features | - | - | 8% | 28% | 58% |

reconstruction, we find the indices of the 50 most important columns and report the proportion of these that represent dense features. For each of the 10 classes (rows of $\mathbf{C}$), we find the indices of the 5 most important columns (features) and compute the proportion of the total of 50 indices that represent dense features. The first row of Table 1 shows the proportion of rows of [$\mathbf{A}$ $\mathbf{B}$] that represent dense features. Comparing this row, respectively to the third and fourth row reveals the importance of $\mathbf{x}$ for reconstruction, and of $\mathbf{u}$ for classification. Indeed, the Acc. and Rec. of Table 1 show that, as the proportion of dense features increases, DenSaE gains reconstruction capability but results in a lower classification accuracy. Moreover, in DenSaE, the most important features in classification are all from $\mathbf{B}$, and the contribution of $\mathbf{A}$ in reconstruction is greater than its percentage in the model, which clearly demonstrates that dense and sparse coding autoencoders balance discriminative and representation power.

Both Tables 1 and 2 show that DenSaE outperforms SCNet$_{\text{LS}}$ in classification and SCNet$_{\text{hyp}}$ in reconstruction. Specifically, for joint training, Table 2 (columns 2,3 and row $J_1$) shows that DenSaE outperforms SCNet$_{\text{hyp}}$ for reconstruction ($32.61 \ll 47.70$) while it is competitive for classification ($98.61 > 98.59$). Moreover, Table 2 (cols 1, 3 and $J_{0.75}$ and $J_1$) highlights that DenSAE has significantly better classification ($98.61 > 96.06$) and reconstruction ($32.61 \ll 71.20$) performances than SCNet$_{\text{LS}}$. Moreover, we observed that in the absence of noise, training SCNet$_{\text{LS}}$ results in dense features with small negative biases, hence, making its performance close to DenSaE with a large number of atoms in $\mathbf{A}$. We see that SCNet$_{\text{LS}}$ in the absence of a supervised classification loss fails to learn discriminative features useful for classification. On the other hand, enforcing sparsity in SCNet$_{\text{hyp}}$ suggests that sparse representations are useful for classification.

**How do reconstruction and classification capabilities change as we vary $\beta$ in joint training?**: In joint training of the autoencoder and the classifier, it is natural to expect that the reconstruction loss should increase compared to disjoint training. This is indeed the case for SCNet$_{\text{LS}}$; as we go from disjoint to joint training and as $\beta$ increases (Table 2), the reconstruction loss increases and classification accuracy has an overall increase. However, for $\beta < 1$, joint training of both networks that enforce some sparsity on their representations, SCNet$_{\text{hyp}}$ and DenSaE, improves reconstruction and classification.

Table 2: DenSaE's performance on MNIST dataset from joint ($J_\beta$) training.

| | | SCNet$_{\text{LS}}$ | SCNet$_{\text{hyp}}$ | $\frac{5\mathbf{A}}{395\mathbf{B}}$ | $\frac{25\mathbf{A}}{375\mathbf{B}}$ | $\frac{200\mathbf{A}}{200\mathbf{B}}$ |
|---|---|---|---|---|---|---|
| | $\frac{\mathbf{A}}{\mathbf{A+B}}$ | - | - | 1.25% | 6.25% | 50% |
| $J_{0.5}$ | Acc. | 96.61% | 97.97% | 97.07% | 97.68% | 96.46% |
| | Rec. | 2.01 | 0.87 | 0.58 | 0.58 | **0.34** |
| $J_{0.75}$ | Acc. | 96.91% | 98.18% | 98.19% | 98.23% | 97.64% |
| | Rec. | 2.17 | 1.24 | 0.75 | 1.11 | 0.51 |
| $J_{0.95}$ | Acc. | 97.23% | 98.51% | 98.23% | 98.44% | 97.81% |
| | Rec. | 4.48 | 1.03 | 1.22 | 1.32 | 0.67 |
| $J_1$ | Acc. | 96.06% | 98.59% | **98.61**% | 98.56% | 98.40% |
| | Rec. | 71.20 | 47.70 | 32.61 | 30.20 | 25.57 |

For purely discriminative training ($\beta = 1$), DenSaE outperforms both SCNet$_{\text{LS}}$ and SCNet$_{\text{hyp}}$ in classification accuracy and representation capability. We speculate that this likely results from the fact that, by construction, the encoder from DenSaE seeks to produce two sets of representations: namely a dense one, mostly important for reconstruction and a sparse one, useful for classification. In some sense, the dense component acts as a prior that promotes good reconstruction.

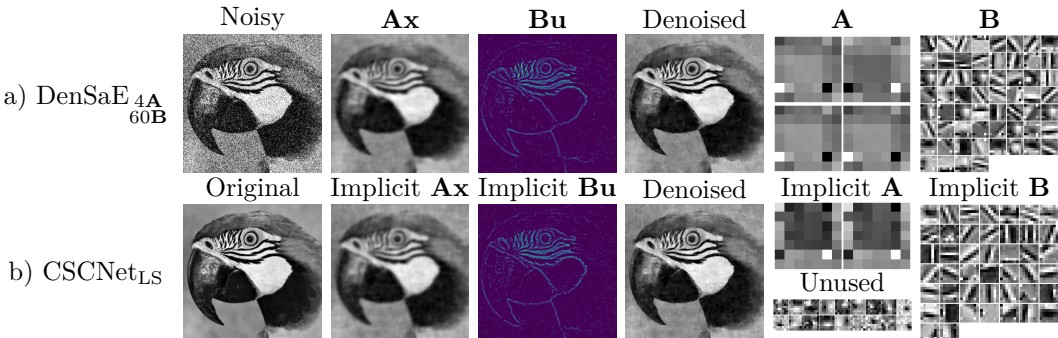

Figure 8: Visualization of a test image for $\tau = 50$. a) DenSaE ($4\mathbf{A}, 60\mathbf{B}$), b) CSCNet$_{\text{LS}}$. $\mathbf{Bu}$ images are scaled and plotted with a different colormap for visualization purposes.

### 5.4.2 Denoising

We trained DenSaE for supervised image denoising when $\beta = 0$ using BSD432 and tested it on BSD68 (Martin et al., 2001). All the networks are trained for 250 epochs using the ADAM optimizer (Kingma, 2014) and the filters are initialized using the random Gaussian distribution. The initial learning rate is set to $10^{-4}$ and then decayed by 0.8 every 50 epochs. We set $\epsilon$ of the optimizer to be $10^{-3}$ for stability. At every iteration, a random patch of size $128 \times 128$ is cropped from the training image and zero-mean Gaussian noise is added to it with the corresponding noise level. We varied the ratio of number of filters in $\mathbf{A}$ and $\mathbf{B}$ as the overall number of filters was kept constant. We evaluate the model in the presence of Gaussian noise with standard deviation of $\tau = \{15, 25, 50, 75\}$.

Table 3: DenSaE's denoising performance on BSD68 as the ratio of filters changes.

| $\tau$ | 1$\mathbf{A}$63$\mathbf{B}$ | 4$\mathbf{A}$60$\mathbf{B}$ | 8$\mathbf{A}$56$\mathbf{B}$ | 16$\mathbf{A}$48$\mathbf{B}$ | 32$\mathbf{A}$32$\mathbf{B}$ |
|---|---|---|---|---|---|
| 15 | **30.21** | 30.18 | 30.18 | 30.14 | 29.89 |
| 25 | **27.70** | **27.70** | 27.65 | 27.56 | 27.26 |
| 50 | **24.81** | **24.81** | 24.43 | 24.44 | 23.68 |
| 75 | 23.31 | **23.33** | 23.09 | 22.09 | 20.09 |

**Ratio of number of filters in A and B**: Unlike reconstruction, Table 3 shows that the smaller the number of filters associated with $\mathbf{A}$, the better DenSaE can denoise images. We hypothesize that this is a consequence of our findings from Section 4 that if the column spaces of $\mathbf{A}$ and $\mathbf{B}$ are suitably unaligned, the easier is the recovery $\mathbf{x}$ and $\mathbf{u}$.

**Dense and sparse vs. sparse coding**: Table 4 shows that DenSaE (best network from Table 3) denoises images better than CSCNet$_{\text{hyp}}$, suggesting that the dense and sparse coding model represents images better than sparse coding.

Table 4: DenSaE vs. CSCNet on BSD68, reporting mean (std).

| $\tau$ | DenSaE | CSCNet$_{\text{hyp}}$ | CSCNet$_{\text{LS}}$ |
|---|---|---|---|
| 15 | 30.21 (1.74) | 30.12 (1.70) | **30.34** (1.79) |
| 25 | 27.70 (1.89) | 27.51 (1.81) | **27.75** (1.89) |
| 50 | **24.81** (1.98) | 24.54 (1.85) | **24.81** (1.97) |
| 75 | **23.33** (1.96) | 22.83 (1.73) | 23.32 (1.95) |

**Dictionary characteristics**: Figure 8(a) shows the decomposition of a noisy test image ($\tau = 50$) by DenSaE. The figure demonstrates that $\mathbf{Ax}$ captures low-frequency content while $\mathbf{Bu}$ captures high-frequency details (edges). This is corroborated by the smoothness of the filters associated with $\mathbf{A}$, and the Gabor-like nature of those associated with $\mathbf{B}$ (Mehrotra et al., 1992). We observed similar performance when we tuned $\lambda_x$, and found that, as $\lambda_x$ decreases, $\mathbf{Ax}$ captures a lower frequencies, and $\mathbf{Bu}$ a broader range.

**CSCNet deviates from sparse coding and implicitly learns Ax + Bu model in the presence of noise**: By training the biases, CSCNet$_{\text{LS}}$ deviates from the sparse coding model; the neural network's bias is directly related to the sparsity enforcing hyper-parameter $\lambda_u$. The larger this bias, the sparser the representations. We observed that CSCNet$_{\text{LS}}$ automatically segments filters into three groups: one with small bias, one with intermediate ones, and a third with large values (see Figure 12). We found that the feature maps associated with the large bias values are all zero. Moreover, the majority of features are associated with intermediate bias values, and are sparse, in contrast to the small

number of feature maps with small bias values, which are dense. We call the dictionary atoms, corresponding such small biases, implicit $\mathbf{A}$. Similarly, dictionary atoms with moderate biases can be seen as implicit $\mathbf{B}$.

These observations suggest that autoencoders implementing the sparse coding model ($\mathbf{y} = \mathbf{Bu}$), when learning the biases by minimizing reconstruction error, implicitly perform two functions. First, they select the optimal number of filters. Second, they partition the filters into two groups: one that yields a dense representation of the input, and another that yields a sparse one. In other words, the architectures trained in this manner *implicitly learn the dense and sparse coding model.* Figure 8(b) shows the filters.

The above-mentioned interpretation of CSCNet with learned biases offers to revisit the optimization-based model used to construct the autoencoder; if the network implicitly learns a bipartite representation, why not explicitly model that structure? Through our experiments, we showed that doing so leads to increased performance with the recovered dictionaries and representations not deviating from their expected behavior. The resulting network is also more interpretable, as we can directly visualize and analyze each component.

# 6 Conclusions

This paper proposed a novel dense and sparse coding model for a flexible representation of a signal as $\mathbf{y} = \mathbf{Ax} + \mathbf{Bu}$. Our first result gives a verifiable condition that guarantees uniqueness of the model. Our second result uses tools from anisotropic compressed sensing to show that, with sufficiently many linear measurements, a convex program with $\ell_1$ and $\ell_2$ regularizations can recover the components $\mathbf{x}$ and $\mathbf{u}$ uniquely with high probability. Numerical experiments on synthetic and real data confirm our observations.

We proposed a dense and sparse autoencoder, DenSaE, tailored to dictionary learning for the $\mathbf{Ax} + \mathbf{Bu}$ model. DenSaE, naturally decomposing signals into low- and high-frequency components, provides a balance between learning dense representations that are useful for reconstruction and discriminative sparse representations. We showed the superiority of DenSaE to sparse autoencoders for data reconstruction and its competitive performance in classification.

### Acknowledgments

Abiy Tasissa acknowledges partial support from the National Science Foundation through grant DMS-2208392. Demba Ba's and Emmanouil Theodosis's work is supported by the National Science Foundation under Cooperative Agreement PHY-2019786 (The NSF AI Institute for Artificial Intelligence and Fundamental Interactions, `http://iaifi.org/`). All the authors would like to thank the the anonymous reviewers for their valuable feedback.

### Author Contributions

Abiy Tasissa, Emmanouil Theodosis and Demba Ba studied the model in (1) and worked on the theoretical analysis presented in Section 4. Emmanouil Theodosis conducted all the experiments in Sections 5.1, 5.2 and 5.3. Bahareh Tolooshams designed and conducted the experiments in Section 5.4 for dictionary learning-based neural architectures. All authors contributed to the writing of the manuscript.

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

## A   Classification and Image Denoising

We warm start the networks by training the autoencoder for 150 epochs using the ADAM optimizer where the weights are initialized with the random Gaussian distribution. Within the network, $\alpha_x$, $\alpha_u$, and $\lambda_u$ are tuned as follows: $\alpha_x$ and $\alpha_u$ are step sizes of the gradient-based iteration; hence, they are chosen to make sure that the recurrence is contractive while they are large enough to make sure gradient information is effective from one another to another. Similarly, $\lambda_u$ is empirically chosen based on the training performance where $lambda_u$ is small enough (to result in non-zero representation), and large enough (to visually observe sparse feature maps). We note that a systematic grid search may improve upon the reported result; however, we suffice to the above-discussed approach as the goal of our paper is a comparative study in a similar training/tuning situation.

The learning rate is set to $10^{-3}$. We set $\epsilon$ of the optimizer to be $10^{-15}$ and used batch size of 16. For disjoint classification training, we trained for 1,000 epochs, and for joint classification training, the network is trained for 500 epochs. All the networks use FISTA (Beck & Teboulle, 2009) within their encoder for faster sparse coding. Table 5 lists the parameters of the different networks. Figure 9 visualizes the most important atoms for reconstruction and classification for disjoint training.

Figure 10 visualizes the reconstruction of MNIST test image for the disjoint training, where the autoencoder is trained for pure reconstruction ($\beta = 0$).

Table 5: Network parameters for MNIST classification experiment.

|  | DenSaE | CSCNet$_{\text{hyp}}$ | CSCNet$_{\text{LS}}$ |
|---|---|---|---|
| # dictionary atoms | | 400 | |
| Image size | | 28×28 | |
| # training examples | | 50,000 MNIST | |
| # validation examples | | 10,000 MNIST | |
| # testing examples | | 10,000 MNIST | |
| # trainable parameters in the autoencoder | 313,600 | 313,600 | 314,000 |
| # trainable parameters in the classifier | 4,010 | 4,010 | 4,010 |
| $\mathcal{S}(.)$ | | ReLU | |
| Encoder layers $T$ | | 15 | |
| $\alpha_u$ | | 0.02 | |
| $\alpha_x$ | 0.02 | - | - |
| $\lambda_u^{\text{init}}$ | 0.5 | 0.5 | 0.0 |

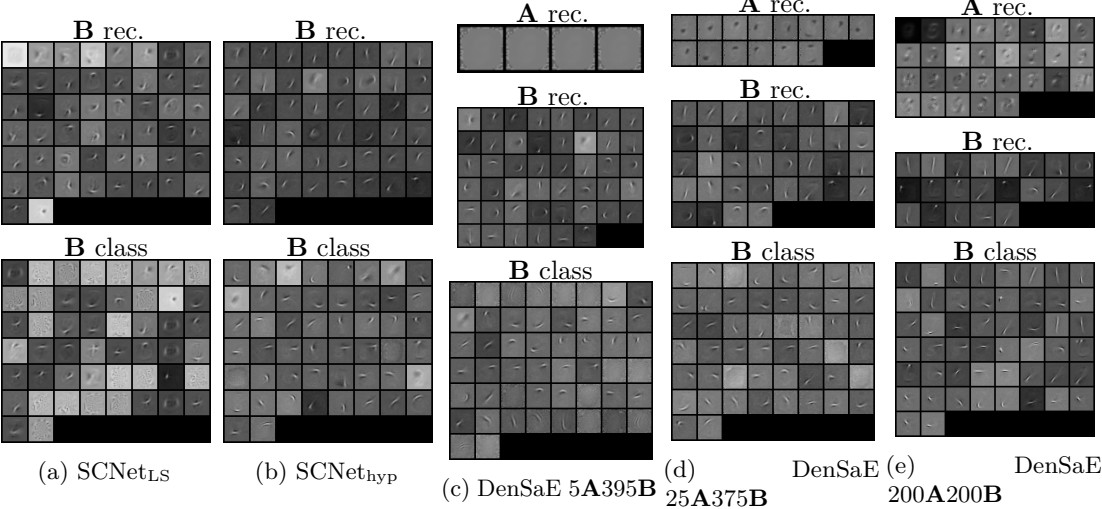

(a) SCNet$_{\text{LS}}$    (b) SCNet$_{\text{hyp}}$    (c) DenSaE 5**A**395**B**    (d) DenSaE 25**A**375**B**    (e) DenSaE 200**A**200**B**

Figure 9: Most important atoms of the dictionary used for reconstruction (rec.) and classification (class.) for disjoint training.

The figure shows that SCNet$_{\text{LS}}$ has the best reconstruction among all, and the second best is DenSaE$_{200\mathbf{A},200\mathbf{B}}$, having the highest number of **A** atoms. Figures 11 visualizes the reconstruction of MNIST test image for the joint training when $\beta = 1$. Notably, in this case, the reconstructions from SCNet$_{\text{LS}}$ do not look like the original image. On the other hand, DenSaE even with $\beta = 1$ is able to reconstruct the image very well. In addition, the figures show how **Ax** and **Bu** are contribution for reconstruction for DenSaE.

We note that as our network is non-convolutional, we do not compare it to the state-of-the-art, a convolutional network. We do not compare our results with the network in Rolfe & LeCun (2013) as that work does not report reconstruction loss and it involves a sparsity enforcing loss that change the learning behaviour.

For denoising, all the trained networks implement FISTA for faster sparse coding. Table 6 lists the parameters of the different networks. Moreover, Figure 12 shows the histogram of learned biases by CSCNet$_{\text{LS}}$ for various noise levels. We note that compared to CSCNet, which has 63K trainable parameters, all the trained networks including CSCNet$^{\text{LS}}$ have 20x fewer trainable parameters. We attribute the difference in performance, compared to the results reported in the CSCNet paper, to this large difference in the number of trainable parameters and the usage of a larger dataset.

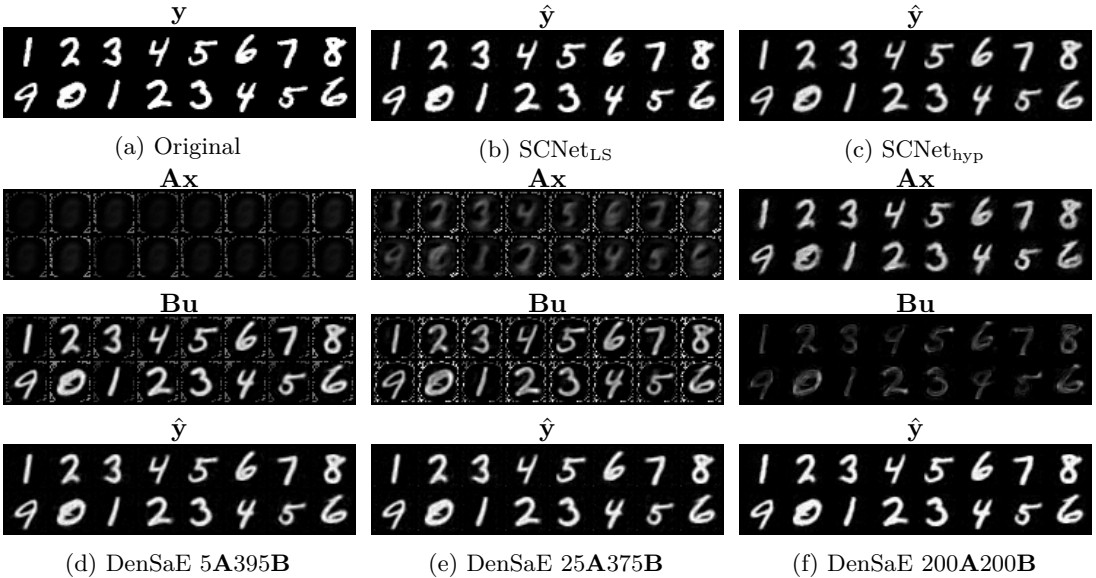

Figure 10: Reconstruction of MNIST test images for disjoint classification.

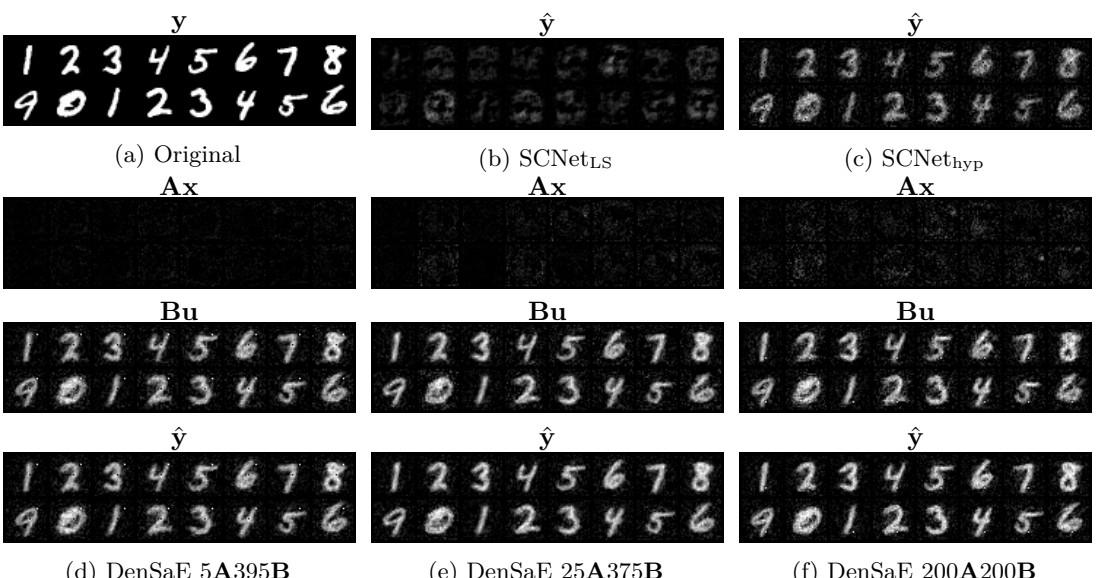

Figure 11: Reconstruction of MNIST test images for joint classification when $\beta = 1$.

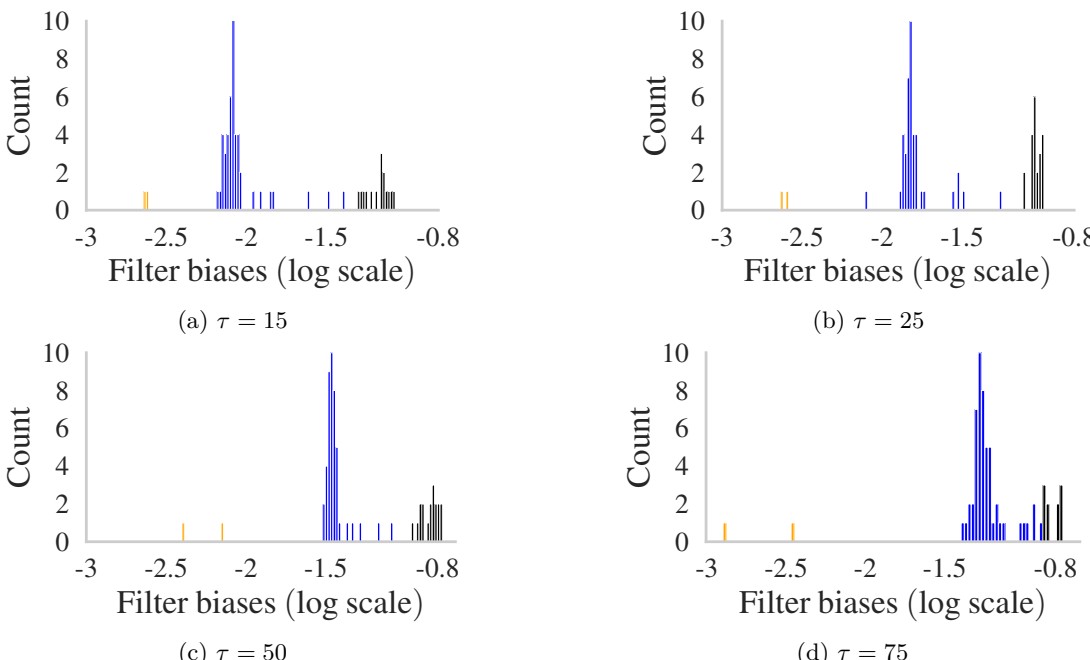

Figure 12: Histogram of biases from CSCNet$_{\text{LS}}$ for various noise levels.

Table 6: Network parameters for natural image denoising experiments.

|  |  | DenSaE | CSCNet$_{\text{hyp}}$ | CSCNet$_{\text{LS}}$ |
|---|---|---|---|---|
| # filters |  | 64 | | |
| Filter size |  | 7×7 | | |
| Strides |  | 5 | | |
| Patch size |  | 128×128 | | |
| # training examples |  | 432 BSD432 | | |
| # testing examples |  | 68 BSD68 | | |
| # trainable parameters |  | 3,136 | 3,136 | 3,200 |
| $\mathcal{S}(.)$ |  | Shrinkage | | |
| Encoder layers $T$ |  | 15 | | |
| $\alpha_u$ |  | 0.1 | | |
| $\alpha_x$ |  | 0.1 | - | - |
| $\lambda_u^{\text{init}}$ | $\tau = 15$ | 0.085 | 0.085 | 0.1 |
|  | $\tau = 25$ | 0.16 | 0.16 | 0.1 |
|  | $\tau = 50$ | 0.36 | 0.36 | 0.1 |
|  | $\tau = 75$ | 0.56 | 0.56 | 0.1 |

# B   Euclidean norm distribution

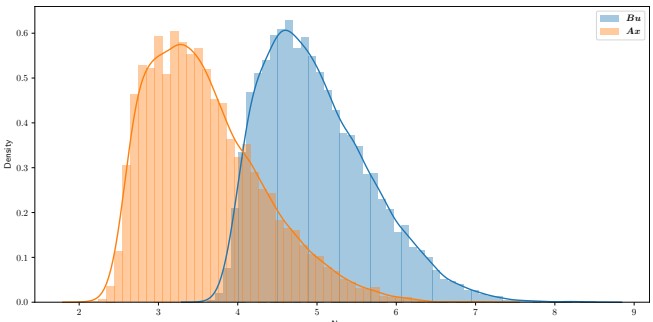

Figure 13: Euclidean norm distribution for the components $\boldsymbol{Ax}$ and $\boldsymbol{Bu}$ of MNIST images.

