# OpenReview forum: "Discriminative reconstruction via simultaneous dense and  sparse coding"
_TMLR — Accepted by TMLR_

### Review · Reviewer_jK8w · 2024-04-23

**Summary Of Contributions:**

The paper tackles the problem of reconstructing two signals, one dense and one sparse, by minimizing a quadratic term for the dense signal and a $\ell_1$ term for the sparse signal, subject to linear constraints.

Theoretical results are proved, which give conditions under which the unique recovery of both signals is possible.

**Audience:**

Yes

**Broader Impact Concerns:**

Broader impact concerns do not apply to the present manuscript.

**Claims And Evidence:**

Yes

**Requested Changes:**

It is important to compare with the naive idea where one writes $y=Ax+bu = [A, B] [x, u]^\top$. In this case, we are given the data matrix $C=[A, B]$ and we know our variable $v:=[x, u ]^\top\in \mathbb{R}^{p+n}$ is $p+k$ sparse. With this, we can apply existing results and obtain conditions that guarantee unique recovery. How do the proposed conditions compare with this approach? Are the proposed conditions better? Such a comparison needs to be included in the revision.

**Strengths And Weaknesses:**

The paper is clearly written, well-organized, and sufficiently reviews prior works.

The experiments are solid and the theoretical results are interesting.

A major weakness is that the theorems seem simple to prove; they are based on basic linear algebraic techniques.

---

> ### Author Response · Authors · 2024-07-09
> **Response to jK8w**
>
> We appreciate the reviewer's feedback on our manuscript. Section 4.1 addresses the Feasibility problem using linear algebra concepts to determine the conditions under which the problem admits a unique solution. Given the fact that the problem is linear, using basic linear algebra techniques was the natural approach. We note that the analysis in 4.2 is not trivial. In particular, standard compressive sensing theory cannot be applied to this setup. Our approach relies on careful construction of the measurement matrices and employs anistropic compressive sensing theory to establish uniqueness with respect to optimality of a mixed objective.
>
> As the reviewer notes, the problem can equivalently be formulated as $y=Ax+bu = [A, B] [x, u]^\top$. Applying standard compressive sensing (CS) results imposes uniform conditions on $[A,B]$, which may lead to sub-optimal recovery guarantees. We will use Theorem 8 to illustrate the difference between the CS approach and ours. For a concrete example, let $A$ be a $20\times 20$ invertible matrix, $B$ be $20\times 80$ matrix and set $k = 2$. To recover $[x\, u]$ exactly, one condition based on CS (see Theorem 1.7 in ref. [1]) is:
> $$
> p+k < \frac{1}{2}\left(1+\frac{1}{\mu[A,B]}\right) \rightarrow k<\frac{1}{2}\left(1+\frac{1}{\mu[A,B]}\right)-p,
> $$ where $\mu[A,B]$ denotes the mutual coherence of the combined dictionary. Note that the range
> of $\mu[A,B]$ is $[\mu_0, 1]$ where $\mu_0$ denotes the Welch bound (see Definition 1.5 in ref. [1]). We make the following observations:
> * The maximum sparsity decreases as $p$ increases. With $p=20$, mutual coherence of the combined
> dictionary needs to be at most $1/43\approx 0.0233$.
> * In our approach, to allow for sparsity $k = 2$, the block coherence needs to be at most
> $1/(2 \sqrt{20})\approx  0.1118$.
> * Theorem 8 relies on block coherence and does not require coherence within the blocks $A$ and $B$. This provides a flexible model, allowing $A$ to be a deterministic coherent dictionary, for example.
>
> Next, consider applying $\ell_1$ minimization and standard compressive sensing theory to guarantee uniqueness given the measurement matrix $H = [A,B]$. One guarantee based on incoherence (see Theorem 1.1 in ref. [2]) states that the number of measurements must be on the order of $\mu_*(p+k)\log(p+n)$, where $\mu_*$ is the coherence of the combined dictionary $H$. In contrast to the mutual coherence mentioned earlier, $\mu_*$ is the smallest number such that the following equality holds for any row of $H$ denoted by $a$:
> $$\max_{1\le t\le (p+n)} |a(i)|^2<\mu_*$$, where $a$ denotes any row of $H$.  It is typically assumed that there is an underlying distribution $F$ from which the rows of $H = [A, B]$ are sampled independently and identically. We note that the range of $\mu_*$,
> for the measurement setup in ref. [2], is $[1,n+p]$. The implication of this is that the sample complexity implicitly requires $\mu = O(1)$. In the case of mixed dictionaries, as in our setup, a coherent matrix $A$ can lead to sub-optimal number of measurements.
>
> In the revised version, we have included a detailed discussion on how our method compares to CS results and have  incorporated the changes requested by the reviewer.
> References
>
> [1] Davenport, Mark A., et al. "Introduction to compressed sensing." (2012): 1-64.
>
> [2] Candes, Emmanuel J., and Yaniv Plan. "A probabilistic and RIPless theory of compressed sensing." IEEE transactions on information theory 57.11 (2011): 7235-7254.

---

### Review · Reviewer_SLyA · 2024-05-01

**Summary Of Contributions:**

The authors focus on the problem of sparse coding and propose a new model for learning to represent a vector $y$ using a sparse representation $u$ based on two dictionaries, $A$ and $B$. The key idea is that $A$ captures low-frequency components of the signal, while $B$ captures high-frequency components. The authors argue that this approach allows for a balance between the model's discriminative and representation power. The authors demonstrate the feasibility of their solution for the proposed sparse coding problem, showing that by using convex optimization, they can recover $x$ and $u$. Finally, they use unrolling to present a dense and sparse autoencoder that demonstrates good representation and discriminative capabilities on the MNIST dataset.

**Audience:**

Yes

**Broader Impact Concerns:**

No concerns

**Claims And Evidence:**

Yes

**Requested Changes:**

Have you considered adding a balancing term between the first and second terms in Eq. 1?

Section 2.1 can you add intuition about applications of compressive sensing from a union of dictionaries? This would help the reader understand the motivation behind this work. Also, in the introduction, it would be useful to present some possible applications for the dense and sparse coding problem.

Section 3: “our the” one of these words should be removed

Section 3.2: Theorem 1 Candes &.. There are two brackets; remove one
Please describe $\beta$ after its first appearance. For example: where $\beta$ is some constant..

Commas missing after equations 5 and 8.


Section 5.1 \sigma is introduced with respect to m,n,p, but those are only mentioned later. This is confusing; please try to improve the presentation of this part to enhance clarity.

CVXY is never introduced.

The last paragraph of section 5.1 writing has some repetition of words such as model and explicitly.

Phase transition is described but please explain that the plot represents the probability of successful recovery (this is not explained).

Figure 2 x-axis overlaps with subfigure names.

Figure 3, the differences are very subtle it is hard to understand the usefulness of the method from this example. Typically, a synthetic example with visualizations should convey the main message of the method clearly.

Section 5.3 third paragraph: “the both the”

Figure 4 shows some differences between Bu and Ax, but again this seems relatively subtle.

Section 5.4 “the parameters .. are tuned via a grid search”. With how many values, what range, and with what objective?

P15. After the questions, the citations for Simon & Elad should be in brackets.

Last paragraph of P15 why are you using << for 32.61 and 47.7 and < for 32.61 and 71.3

P16 What are the train test splits for these experiments? Are they performed several times?

On the same page what is $\beta$ please refer to the place it is described.

Table 4 without standard deviations, it is hard to compare these results; both DenSaE and CSCNet_{ls} seem to obtain similar results.

**Strengths And Weaknesses:**

**Strengths**

Overall, the paper is well written and easy to follow.

The authors present a new sparse coding problem that uses two types of dictionaries and learns sparse and dense components.

They provide theoretical analysis and describe conditions for unique solutions.

Synthetic data is used to validate that the solution can be recovered with high probability.

**Weaknesses**

The motivation for the proposed model is not really clear. I do understand the importance of sparse coding and its multiple use cases in denoising images and audio signals. However, I do not fully understand what advantages are gained from the proposed setting. Specifically, in what cases is it important to balance between representation and discriminative power?

On the same note, the applications are semi-synthetic. What real-world problem should the reader be using this model for?

Overall, the proposed model's performance on real data seems comparable to the existing SCNet, and the authors have not highlighted which data regime their method is beneficial for and vice versa.

---

> ### Author Response · Authors · 2024-07-09
> **Response to SLyA**
>
> First, we would like to thank the reviewer for the feedback on our manuscript.
>
> **Motivation and applications**: Combinations of sparse and smooth signals are of interest in anomaly detection and robustness. Consider the decomposition of an m-dimensional signal $y$ into three components: $y = y_1 + y_2+n$ where $y_1=Ax$ is the smooth part, $y_2=Bu$ is structured noise and $n$ is noise (e.g., Gaussian noise). Such a decomposition has been studied in applications such as manufacturing and anomaly detection in images. Our model uses an overcomplete $B$, and the DenSaE algorithm also facilitates learning the dictionaries $A$ and $B$ in cases where there is no prior knowledge or model for the noise.
>
> **Advantages of proposed model**: Learning representations are key to intelligence systems; in real-world applications, DenSaE offers a framework to learn a structured representation that can be used and analyzed for interpretability analysis.
>
> ### Requested changes
> ----------------------------------------
> The revised version addresses all the changes requested by the reviewer. We have highlighted some of the answers below:
>
> **Question**: Differences between $Bu$ and $Ax$.
>
> **Answer 1**: Figure 3 highlights the decomposition of actual data under the proposed optimization. The goal of the figure is to give a concrete example of the outputs of the optimization (the vectors $x$ and $u$) and how they are used to reconstruct the input completely. The optimization of (26) minimizes three terms: the reconstruction term, the smoothness term $\lVert Ax \rVert_2^2$, and the sparsity term $\lVert u\rVert_1$. In Figure 3 we can see that the reconstruction term is indeed minimized (as the leftmost part resembles the actual input without noise) and the sparsity of $u$ is also enforced, as there are very few nonzero elements. Finally, (26) is also penalizing smoothness, which is subtly shown in Figure 3 when one compares the terms $Bu$ and $Ax$. Precisely because in Figure 3 the difference between $Ax$ and $Bu$ is subtle, we quantitatively evaluated the difference in smoothness between $Ax$ and $Bu$ in Figure 4. In Figure 4 the distributions of $Ax$ and $Bu$ have distinct means (with over 40 units of total variation in difference), indicating the success of the optimization in (26). Note that (26) does not enforce any distributional changes between $Ax$ and $Bu$, but only penalizes $\lVert Ax\rVert_2^2$.
>
> **Question**: Tuning of parameters in Section 5.4.
>
> **Answer**: We would like to clarify that $\alpha_x$, $\alpha_u$, and $\lambda_u$ are tuned as explained below; we have revised the text from "grid search" to "tuning" to reflect our strategy. $\alpha_x$ and $\alpha_u$ are step sizes of the gradient-based iteration; hence, they are chosen to make sure that the recurrence is contractive while they are large enough to make sure gradient information is effective from one another to another. Similarly, $\lambda_u$ is empirically chosen based on the training performance where $\lambda_u$ is small enough (to result in non-zero representation), and large enough (to visually observe sparse feature maps). We note that a systematic grid search may improve upon the reported result; however, we suffice to the above-discussed approach as the goal of our paper is a comparative study in a similar training/tuning situation.
>
> **Question**: P16 What are the train test splits for these experiments? Are they performed several times?
>
> **Answer**: For MNIST, there is a 10K test example. In addition, we split the 60K train data into 50 K training and 10 K validation; the validation is used to choose the best-performing results. For Image denoising, we use CBSD (colored one) for training, and there is a BSD68 set of 68 images that is commonly used for testing (the network has never seen this test set of images). Table 5 and 6 in the Appendix summarize this information. We have not performed it several times.
>
> **Question**: Standard deviations absent in table 4 and comparison of DenSaE and $\text{CSCNet}_{ls}$.
>
> **Answer**: We have now added standard deviations in Table 4. SCNet starts with a sparse coding model and claims to learn sparse representations; however, it treats the sparsity-inducing parameter (the bias of the neural network) as a learnable parameter. This learnable modification of a sparse coding-based network has the following consequence (this is the interpretation of our paper to SCNet): SCNet with learnable bias deviates from sparse coding to sparse + dense coding. Indeed, such deviation is one main reason for their framework to perform better than sparse coding-based networks with fixed regularization parameters. Moreover, this outperformance supports our argument that sparse + dense model is superior to a sparse model. Overall, SCNet with learnable bias does dense + sparse coding implicitly, and this is why SCNet performs similarly to DenSaE, which explicitly does dense + sparse coding.

---

> > ### Comment · Reviewer_SLyA · 2024-07-25
> > **Response to authors**
> >
> > I thank the authors for their response to my comments.
> > Most of my concerns have been addressed by the authors.
> > The new approach and theoretical analysis are the main strengths of the paper.
> > The added standard deviation for Table 4 suggests that all methods are within the standard deviation of one another in this specific example.
> > Therefore, as another reviewer noted, the paper's main weaknesses remain in its experimental and application parts.

---

### Review · Reviewer_R52K · 2024-06-25

**Summary Of Contributions:**

This work examines a signal recovery where given a measurement y the goal is to recover a sparse vector u and a dense vector x as sensed through two separate matrices.
The authors provide criteria on the matrix and signal under which given vectors u and v are unique solutions under the optimization problem and provide an algorithm to recover them via convex optimization.
In their experiments, they show the phase transitions for perfect recovery in terms of the ratios of sparsity, number of measurements and the dimension of the original vectors.
They compare how their novel autoencoder architecture, which is informed by their previous analysis to simultaneously optimize a sparse and a dense representation, compares to only sparse approaches in the case of MNIST images.
While their model is better in many case, they reveal an interesting connection between the two models and show that the sparse model implicitly follows a similar approach as the dense model.

**Audience:**

Yes

**Broader Impact Concerns:**

I have no concerns.

**Claims And Evidence:**

Yes

**Requested Changes:**

### Up to the authors
The empirical analysis would be nicely complemented by a supplementary analysis of a dataset that exhibits different levels of supposed sparsity than BDS68. It would be also nice to understand that (e.g. more sparse) types of noise or corruption influence the learned dictionaries in a predicable way.

### Form - should definitely be incorporated
- all over the manuscript \citep and \cite were not used appropriately
- Sec.2.2, first sentence, is -> in
- Sec.2.2., fat is not clear from context
- Sec.2.3., first sentence, do you mean u instead of y?
- End Proof Thm 4 -> follows
- Fig.1 label color bar
- Sec.5.3. "In order to further validate ..." -> one 'the' too many
- Sec.5.3. sprase -> sparse
- (27)-(29) define the size of X, U, what is the range of l?
- (27) why do the matrices do not have a l-index?
- Sec.5.4. (i) filters is used the first time -- what exactly is meant?
- Sec 5.3. last sentence -> missing a "the"
- Sec 5.4.2 (see ??? 10)

**Strengths And Weaknesses:**

### Strengths
- The writing style is clear and calm. The authors take great care and time to introduce the context and related works.
- The theoretical results are interesting, the overlay of dense and sparse features is natural and it is interesting to see how this case comparse to the separate cases.
- The empirical results compare the performance of two different architectures of autoencoders, informed by different principles, in depth. Notably they go beyond this by introspecting the models and giving intuitive explanations for the different/similar performances.

### Weaknesses
- The empirical evaluation is not comprehensive in terms of the variety of real-world data explored.

---

> ### Author Response · Authors · 2024-07-09
> **Response to R52K**
>
> We appreciate the reviewer's feedback on our work. We have incorporated all the requested changes in the revised version. In future work, we plan to do comprehensive datasets and explore different noise models.
>
> For the color bar, we clarified in the text that Figure 3 shows the probability of successful recovery (as was also requested by another reviewer) and we expanded the caption of the figure to better explain the color bar.

---

> > ### Comment · Reviewer_R52K · 2024-07-19
> > **Response**
> >
> > Thank you for resolving the comments!

---

### Decision · Action_Editor_gPGe · 2024-08-07

**Recommendation:** Accept as is

**Comment:**

Please consider improving your experimental section according to the reviewers' comments.

**Audience:**

All reviewers agree that there is audience for this paper in TMLR.

**Claims And Evidence:**

All reviewers were satisfied with the authors' responses, and agreed that overall the revised paper meets the expectations of evidence to the claims made in the paper.

Still, it was noted that the weaker part of the paper is the experimental section, and while there is a concuss to accept the paper, I encourage the authors to read the reviews again and try to improve their experimental section accordingly.